



# The role of boundary layer processes in summer-time Arctic cyclones

Hannah L. Croad[1], John Methven[1], Ben Harvey[1, 2], Sarah P. E. Keeley[3], and Ambrogio Volonté[1]

[1]Department of Meteorology, University of Reading, Reading, UK
[2]National Centre for Atmospheric Science, University of Reading, Reading, UK
[3]European Centre for Medium-Range Weather Forecasts (ECMWF), Reading, RG2 9AX, UK

**Correspondence:** Hannah Croad (h.croad@pgr.reading.ac.uk)

**Abstract.** Arctic cyclones are the most energetic weather systems in the Arctic, producing strong winds and precipitation that present major weather hazards. In summer, when the sea ice cover is reduced and more mobile, Arctic cyclones can have large impacts on ocean waves and sea ice. While the development of mid-latitude cyclones is known to be dependent on boundary layer (BL) turbulent fluxes, the dynamics of summer-time Arctic cyclones and their dependence on surface exchange

processes have not been investigated. The purpose of this study is to characterise the BL processes acting in summer-time Arctic cyclones and understand their influence on cyclone evolution. The study focuses on two cyclone case studies, each characterised by a different structure during growth in the Arctic: (A) low-level dominant vorticity (warm-core) structure, and (B) upper-level dominant vorticity (cold-core) structure, linked with a tropopause polar vortex. A potential vorticity (PV) framework is used to diagnose the BL processes in model runs from the ECMWF Integrated Forecasting System model.

Both cyclones are associated with frictional Ekman pumping, and downward sensible heat fluxes over sea ice. However, the frictional baroclinic PV generation process (governed by the angle between the lower tropospheric thermal wind vector and the surface wind) acts differently in A and B due to the different cyclone structures. Positive PV is generated in A around the bent-back warm front, like in typical mid-latitude cyclones. However, the same process produces negative PV tendencies in B, shown to be a consequence of the vertically-aligned columnar vortex structure. This frictional process also acts to cool the

lower troposphere, reducing the warm-core anomaly in A, and amplifying the cold-core anomaly in B. Both cyclones attain a vertically-aligned cold-core structure that persists for several days after maximum intensity, which is consistent with cooling from frictional Ekman pumping, frictional baroclinic PV generation, and downward sensible heat fluxes. This may help to explain the longevity of isolated cold-core Arctic cyclones with columnar vorticity structure.

## 1  Introduction

The rapid loss of sea ice due to anthropogenic global warming (e.g. Comiso, 2012; Meier et al., 2014) is permitting human activity to expand into the summer-time Arctic. For example, reduced sea ice extent and thickness will open up shorter shipping routes through the Arctic between Atlantic and Pacific ports (Melia et al., 2016). This human activity will be exposed to the risks of Arctic weather during the summer, including Arctic cyclones.



Arctic cyclones are synoptic-scale low pressure systems developing, or moving into, the Arctic. They produce some of the most impactful weather in the Arctic, with strong winds at the surface and sometimes extreme ocean waves (e.g. Thomson and Rogers, 2014; Waseda et al., 2018). Arctic cyclones are also associated with atmospheric forcings that have large impacts on the sea ice (e.g. Simmonds and Keay, 2009; Asplin et al., 2012; Zhang et al., 2013; Peng et al., 2021). As the climate warms the summer-time Arctic is becoming increasingly dominated by the marginal ice zone (MIZ; Strong and Rigor, 2013), a band of

fragmented ice floes separating the ice-free ocean and the main ice pack. In recent years the MIZ has widened by 39% (Strong and Rigor, 2013) in summer, with MIZ fraction (MIZ extent divided by total sea ice extent) increasing by more than 50% (Rolph et al., 2020). Thinner and more mobile ice in the MIZ will result in enhanced surface interactions with Arctic cyclones, with greater surface drag due to ice floe morphology (Lüpkes and Birnbaum, 2005; Elvidge et al., 2016), and enhanced surface sensible and latent heat fluxes due to a greater exposure of the ocean surface. For example, it has been argued that record-low

sea ice extent in 2012 was exacerbated by an extremely strong cyclone, the great Arctic cyclone of 2012 (AC12; Simmonds and Rudeva, 2012), with enhanced ice melt due to increased upward ocean heat transport (Zhang et al., 2013).

Forecast skill in the Arctic is lower, but generally comparable, to that in northern hemisphere mid-latitudes (Jung and Matsueda, 2016; Sandu and Bauer, 2018), based on 500 hPa geopotential anomaly correlation scores. Lower predictability in the

Arctic is likely related to the relative sparsity of observations there, resulting in larger uncertainties in initial conditions for numerical weather prediction models. Previous work has also demonstrated that the forecast skill of Arctic cyclones is lower than that of mid-latitude cyclones. Yamagami et al. (2018a) demonstrated that the mean predictability of 10 extraordinary Arctic cyclone cases was 2.5–4.5 days, around 1–2 days less than that of northern hemisphere mid-latitude cyclones (Froude, 2010). Furthermore, using ensemble forecasts, Capute and Torn (2021) demonstrated that the ensemble mean root-mean-square error

and ensemble standard deviation for cyclone position was higher for 100 selected summer-time Arctic cyclones, than for 89 selected winter-time Atlantic mid-latitude cyclones. Yamagami et al. (2018b) examined the predictability of AC12 and found that the position variability was greater than intensity variability between ensemble forecasts. Furthermore, ensemble members that best captured the upper-level vortex merger associated with AC12 produced the best forecasts, demonstrating that an understanding of cyclone dynamics and mechanisms is critical for predictability. Improvements in Arctic cyclone forecasting can

likely be achieved through a better understanding of the physical processes that distinguish Arctic cyclones from mid-latitude cyclones, including the different growth mechanisms and interaction with sea ice.

Vessey et al. (2022) demonstrated that the composite structure of intense summer-time Arctic cyclones is distinct to that of intense winter-time Arctic and North Atlantic mid-latitude cyclones. Summer-time Arctic cyclones undergo a structural tran-

sition at the time of maximum intensity, from a tilted baroclinic system to an axisymmetric cold-core structure. The mean lifetime of the summer-time Arctic cyclones was also found to be more than 3 days greater than that of winter-time Arctic cyclones, and 4 days greater than that of winter-time North Atlantic mid-latitude cyclones (Vessey et al., 2022). The longevity of some Arctic cyclones and the transition to an axisymmetric cold-core structure has also been documented in several case studies (e.g. Simmonds and Rudeva, 2012; Tanaka et al., 2012; Aizawa and Tanaka, 2016; Yamagami et al., 2017; Tao et al.,



60   2017).

Many of the growth mechanisms of summer-time Arctic cyclones are the same for mid-latitude cyclones, such as baroclinic instability and lee cyclogenesis. However, sustained cyclone interaction with tropopause polar vortices (TPVs), long-lived vortices on the tropopause with horizontal scales of less than 1500 km (Cavallo and Hakim, 2009), is a characteristic of the Arctic

(where there is typically an absence of a strong zonal jet stream in the upper troposphere). Therefore a possible classification scheme for summer-time Arctic cyclones is to consider whether or not they develop with a TPV. Accordingly, Gray et al. (2021) refer to Arctic cyclones that are (i) "unmatched" and (ii) "matched" with a TPV during development. The authors use a statistical matching criterion based on a threshold distance between tracked TPVs and low-level cyclones to classify whether a cyclone is matched or not. It was found that unmatched cyclones are initially dominated by low-level vorticity, developing

as part of a baroclinic wave. In these cyclones, vorticity decreases with height, and therefore they have low-level warm-cores (a horizontal temperature maximum) by thermal wind balance. These cyclones occur most commonly along the northern coast of Eurasia (Fig. 7 in Gray et al. (2021)), in association with baroclinicity on the Arctic Frontal Zone (AFZ), where there is a strong meridional temperature gradient in summer due to a land-sea temperature contrast (Serreze et al., 2001; Day and Hodges, 2018). In contrast, matched cyclones are dominated by upper-level vorticity (with tropospheric cold-cores by thermal

wind balance). Matched cyclones are associated with reduced tilt, suggesting reduced baroclinicity, and a single columnar vortex structure at maximum intensity (like the summer-time Arctic cyclone composite in Vessey et al. (2022)). Matched cyclones track most frequently along the North American coastline (Fig. 7 in Gray et al. (2021)), consistent with the climatological location of TPVs (Cavallo and Hakim, 2010). In this study Arctic cyclones will be classified in terms of their vorticity structure during development, as either (i) low-level dominant, or (ii) upper-level dominant. This is similar to the unmatched and

matched classification used by Gray et al. (2021), but focuses on the cyclone structure itself rather than the identification of TPVs.

One of the biggest uncertainties in the modelling of Arctic cyclones is the interaction with the surface and sea ice. Turbulent fluxes of momentum, heat and moisture in the boundary layer (BL) are critical to understanding the cyclone-sea ice interaction.

It is known that BL turbulent fluxes have large impacts on the evolution of mid-latitude cyclones. For example, friction acts to reduce the intensity of cyclones, with Valdes and Hoskins (1988) demonstrating that surface drag can reduce the growth rates of baroclinic systems by up to 50%. The dominant physical mechanism responsible for this is often assumed to be Ekman pumping. In Ekman pumping, BL friction causes the near-surface wind to weaken and turn toward the low centre. The subsequent convergence forces ascent at the BL top, which acts to spin down the cyclone via barotropic vortex squashing.


Previous studies have used a potential vorticity (PV) framework to identify the mechanisms by which BL processes impact mid-latitude cyclones. PV is a central variable in the evolution of baroclinic systems (e.g. Hoskins et al., 1985), considering





both vorticity and stratification:

$$P = \frac{1}{\rho} \boldsymbol{\zeta_a} \cdot \nabla \theta \tag{1}$$

where $P$ = Rossby-Ertel PV (K m$^2$ kg$^{-1}$ s$^{-1}$), $\rho$ = density (kg m$^{-3}$), $\boldsymbol{\zeta_a}$ = absolute vorticity (s$^{-1}$), and $\theta$ = potential temperature (K). PV is materially conserved for adiabatic, inviscid motion, but not in the presence of friction and diabatic heating. Lagrangian changes of PV are expected in the BL where friction and diabatic heating are important.

In the PV framework, when friction weakens the near-surface winds in a cyclone it reduces the azimuthal cyclonic circulation (i.e. the vertical component of vorticity), and therefore reduces PV in the BL near the low centre. In a balanced state, there is both a reduction in cyclonic circulation and BL static stability. To achieve this, isentropes must rise, increasing the static stability above the BL. To conserve PV above the BL, vorticity must decrease there. This is the Ekman pumping process, characterised by a negative PV tendency in the BL, and a secondary circulation with inflow within the BL and ascent near the cyclone centre which acts to spin-down vorticity in and above the BL.

The PV framework also reveals a second frictional process in mid-latitude cyclones, in which friction acts to alter horizontal vorticity and circulation in the x-z and y-z planes. This process is called "frictional baroclinic PV generation" and is most prominent in regions of strong horizontal temperature gradients, such as fronts. In this process, PV is generated in the BL where surface winds oppose the tropospheric thermal wind (Cooper et al., 1992). This frictional baroclinic PV generation occurs mainly to the east and north-east of cyclone centres along the warm front (Stoelinga, 1996; Adamson et al., 2006; Plant and Belcher, 2007; Vannière et al., 2016). Adamson et al. (2006) found evidence of both frictional Ekman pumping and baroclinic PV generation in their idealized life-cycle model, but showed that baroclinic generation dominated. Consistent with this, Stoelinga (1996) found that BL friction generated mainly positive low-level PV in their mid-latitude cyclone modelling study. However, a sensitivity experiment showed that the overall effect of surface drag was to produce a weaker cyclone due to the reduced development of the upper-level wave (i.e. reduced baroclinicity and mutual growth of the upper and lower waves). This is consistent with the baroclinic PV mechanism described by Adamson et al. (2006), whereby positive PV generated in the BL is ventilated out of the BL by the warm conveyor belt (WCB), and advected above the low centre. This positive PV is associated with increased static stability above the BL, acting as an insulator to reduce the coupling of the upper and lower levels, reducing the cyclone growth rate. Note that both frictional Ekman pumping and the baroclinic PV mechanism have impacts above the BL (in fact Boutle et al. (2015) suggest that these processes act in union to maximize cyclone spin-down), demonstrating that the role of surface friction in a cyclone is more complicated than simple Ekman spin-down of vorticity in a barotropic vortex.

Sensible heat fluxes have a direct effect on PV by altering static stability in the BL. For example, Chagnon et al. (2013) identified a region of negative BL PV behind the cold front of a mid-latitude cyclone, generated due to strong upward sensible heat fluxes (i.e. the surface losing heat to the overlying atmosphere), associated with reduced BL static stability. Sensible heat





fluxes also have indirect effects, modifying the action of frictional processes by altering BL stability and by weakening frontal gradients (Plant and Belcher, 2007).

This PV framework has been used exclusively to study mid-latitude cyclones. The role of BL processes in the evolution of summer-time Arctic cyclones has not yet been investigated. Differences are expected for several reasons; (i) the sea ice surface in summer is characterised by increased surface roughness in the MIZ and variable sensible heat fluxes, (ii) there are longer-lived cyclones in the summer-time Arctic, so BL processes have a longer time to act, and (iii) there are different cyclone growth mechanisms and structures in the summer-time Arctic, such that BL processes might impact cyclone evolution in dif-
ferent ways.

In this study we aim to answer the following questions using two case studies from summer 2020:

1. What is the nature of the BL processes acting in contrasting summer-time Arctic cyclones?

2. How does the nature of the BL processes change as the cyclones evolve?

3. How do these compare with mid-latitude cyclones?

4. What is the impact of the BL processes on Arctic cyclones outside the BL?

The paper is structured as follows. The methodology is described in Section 2, including the model setup employed and details of the PV framework. Section 3 describes two Arctic cyclone case studies from summer 2020. The main results are presented in Section 4, with a more general discussion in Section 5. The study is concluded in Section 6.

## 145 2 Methodology

### 2.1 Reanalysis data

The study uses data from the ERA5 dataset, the fifth-generation European Centre for Medium-Range Weather Forecasts (ECMWF) reanalysis product (Hersbach et al., 2020). ERA5 was produced using the ECMWF's Integrated Forecasting System (IFS) model cycle 41r2, which was operational from 8 May to 21 November 2016. The model has spectral truncation TL639
(horizontal resolution ∼31 km), and 137 terrain-following hybrid-pressure levels from the surface to 0.01 hPa. Six-hourly data on a 0.25° regular latitude-longitude grid is used to perform an analysis of Arctic cyclones from the 2020 extended summer (May-September) season.

### 2.2 IFS model runs

The primary tool used in the study is the ECMWF global IFS model, coupled with dynamic ocean and sea ice models. Forecasts
were run using IFS model cycle 47r1, with spectral truncation O640 (horizontal resolution ∼18 km), and 91 terrain-following





hybrid pressure levels up to 0.01 hPa. This is the same setup as the control member of the ECMWF ensemble forecasting system (ENS) which was operational from 30 June 2020 to 10 May 2021. A prognostic dynamic-thermodynamic sea ice model, the Louvain-la-Neuve Ice Model (LIM version 2), is used, incorporated into the dynamical ocean model (NEMO version 3.4; Nucleus for European Modelling of the Ocean).

### 2.3 Arctic cyclone tracking

Tracks of Arctic cyclones are identified from ECMWF ERA5 reanalysis data and from the ECMWF IFS model runs (the control member of ENS, model cycle 47r1) using the TRACK programme developed by Hodges (1994, 1995). The TRACK algorithm is employed on the T5-63 and T5-42 filtered 850 hPa relative vorticity from ERA5 and the ENS control member respectively, identifying anomalies exceeding $10^{-5}$ s$^{-1}$. Only tracks that last longer than 1 day and travel more than 1000 km are retained.

### 2.4 A modified cyclone phase space

A modified cyclone phase space for characterising the structure of Arctic cyclones is proposed. This phase space is based on the thermal asymmetry and thermal wind structure of a cyclone, as in Hart (2003), but is presented in a non-dimensionalised and more direct way. Thermal asymmetry is quantified here as a non-dimensionalised depth-integrated baroclinicity, B, over the 925–700 hPa layer (above the BL but below the 'steering' level). As in Hart (2003), this represents the linear variation of temperature across the cyclone (of radius 500 km), by splitting the cyclone into a right (R) and left (L) half:

$$B = \frac{1}{f_0 L N} \frac{g}{\theta_0} \frac{1}{\Delta p} \int\limits_{700hPa}^{925hPa} (\theta_R - \theta_L) \, dp \qquad (2)$$

where $f_0$ = Coriolis parameter (s$^{-1}$), $L$ = cyclone length scale (500 km), $N$ = Brunt–Väisälä frequency (0.01 $s^{-1}$), $g$ = gravitational acceleration (9.81 ms$^{-2}$), $\theta_0$ = reference potential temperature (273 K), $p$ = pressure (hPa), $\theta_R$ and $\theta_L$ = areal mean potential temperature over a semi-circle of radius 500 km to the right and left of the cyclone (K). In the Hart (2003) phase space, the cyclone is split in half by the cyclone motion vector. However, Arctic cyclones can be associated with slow movement and remain quasi-stationary for considerable periods of time, such that the motion vector is not well defined. Hence, here B is calculated at every 10° bearing, with the maximum value of B being used at each time. The larger the value of B, the greater the asymmetry and baroclinicity of a cyclone.

Thermal wind balance can be written in terms of vorticity (Equation 12.6 in Hoskins and James (2014)):

$$\frac{\partial \xi}{\partial z} = \frac{1}{f_0} \nabla^2 b' \qquad (3)$$





where $\xi$ = relative vorticity (s$^{-1}$), $z$ = height (m), $b'$ = buoyancy anomaly ($\frac{g}{\theta_0}\theta'$, ms$^{-2}$), and $\theta'$ = potential temperature anomaly. A system in the northern hemisphere where $\xi$ increases with height ($\frac{\partial\xi}{\partial z} > 0$; upper-level dominant) must be in balance with a cold-core thermal wind structure (negative buoyancy anomaly), as $\nabla^2 b' > 0$ corresponds to $b' < 0$ (since $\nabla^2 b' \sim -\frac{b'}{L^2}$ for systems of length scale $L$). In contrast, $\xi$ decreases with height (low-level dominant) in warm-core systems. Hence, the thermal wind structure is quantified here as a non-dimensionalised vertical gradient of relative vorticity in the 700–400 hPa layer (above the 'steering' level but below the tropopause):

$$Ro_T = -\frac{L}{N}\frac{\partial\xi}{\partial z} \tag{4}$$

where $\frac{\partial\xi}{\partial z}$ is calculated by a linear regression fit of $\xi$ at 50 hPa intervals between 700 hPa and 400 hPa. The quantity $Ro_T$ is the thermal Rossby number; the non-dimensional ratio of the inertial force due to the thermal wind and the Coriolis force. The form in (4) is obtained using the Burger number ($Bu = \frac{NH}{f_0 L}$ where $H$ is the height scale), the non-dimensional ratio of the density stratification and Earth's rotation in the vertical, which is assumed to be 1 for the synoptic scale. A positive $Ro_T$ indicates a low-level dominant cyclone and therefore a warm-core structure, whilst a negative $Ro_T$ corresponds to an upper-level dominant or cold-core structure.

The circularly symmetric component of $\theta'$ (and equivalently $b'$) can be expressed in terms of the potential temperature at the cyclone centre, $\theta_C$, and a background potential temperature $\theta_B$ (representing the average value at a 500 km radius). Making a second-order finite difference approximation: $\nabla^2\theta' \approx -\frac{\theta_C - \theta_B}{L^2}$, and substituting into (4) using thermal wind balance in (3) gives:

$$Ro_T = \frac{1}{f_0 L N}\frac{g}{\theta_0}(\theta_C - \theta_B) \tag{5}$$

The appeal of this cyclone phase space is that it is non-dimensionalised, and it is dependent on the potential temperature structure of the cyclone only. Note that in (2) and (5), the quantities $\theta_R - \theta_L$ and $\theta_C - \theta_B$ are scaled in the same way, such that their magnitudes can be directly compared.

## 2.5 PV framework

In the presence of friction and diabatic heating, the Lagrangian evolution of PV is given by:

$$\frac{DP}{Dt} = -\frac{1}{\rho}\nabla \cdot \boldsymbol{J} = \frac{1}{\rho}\left[(\nabla \times \boldsymbol{F})\cdot\nabla\theta + \boldsymbol{\zeta_a}\cdot\nabla\left(\frac{D\theta}{Dt}\right)\right] \tag{6}$$

where $\rho$ = density (kg m$^{-3}$) and $\boldsymbol{F}$ = frictional force vector (m s$^{-2}$). $\boldsymbol{J} = -\boldsymbol{F} \times \nabla\theta - \boldsymbol{\zeta_a}\frac{D\theta}{Dt}$ is the PV flux arising from non-conservative terms in the Haynes and McIntyre (1987) form. The first term on the RHS of (6) represents frictional effects on PV, whilst the second term represents diabatic heating effects, which can be split into sensible ($sen$) and latent ($lat$) heat flux





contributions. Application of (6) in the BL would require full three-dimensional fields of friction and diabatic heating, which would be strongly dependent on the three-dimensional structure of parameterised tendencies and would be difficult to interpret. Therefore a simplified expression for the BL depth-averaged PV tendency was derived by Cooper et al. (1992). It is assumed that the horizontal variation of fluxes is substantially smaller than the vertical variation in the BL such that:

$$\boldsymbol{F} = \frac{1}{\rho}\frac{\partial \boldsymbol{\tau}}{\partial z}, \; \frac{D\theta}{Dt}\bigg|_{sen} = -\frac{\partial H}{\partial z} \tag{7}$$

where $\tau$ is the momentum flux and $H$ is the sensible heat flux. A linear flux gradient is also assumed in the BL, such that fluxes can be specified as a product of their surface values, $\boldsymbol{\tau}_S$ and $H_S$, decreasing linearly to zero at the top of the boundary layer with height $h$:

$$\boldsymbol{\tau} = \boldsymbol{\tau_S}S(z), \; H = H_S S(z), \; S(z) = \left(1 - \frac{z}{h}\right) \tag{8}$$

where $S(Z)$ is a linear function of height in the BL. Note that the convention used here is that $\boldsymbol{\tau}_S$ is taken to be in the same direction as the surface wind (i.e. $\boldsymbol{\tau}_S$ is the stress that the atmosphere exerts on the surface). The frictional and sensible heating terms on the RHS of (6) can be decomposed into the contributions from the vertical and horizontal components in the dot products, with subscripts $(V)$ and $(H)$ respectively:

$$\frac{DP}{Dt} = \underbrace{\frac{1}{\rho_0}[(\nabla \times \boldsymbol{F}) \cdot \nabla \theta]_V}_{(F_{EK})} + \underbrace{\frac{1}{\rho_0}[(\nabla \times \boldsymbol{F}) \cdot \nabla \theta]_H}_{(F_{BG})} + \underbrace{\frac{1}{\rho_0}\left[\boldsymbol{\zeta_a} \cdot \nabla\left(\frac{D\theta}{Dt}\right)_{sen}\right]_V}_{(S_V)}$$
$$+ \underbrace{\frac{1}{\rho_0}\left[\boldsymbol{\zeta_a} \cdot \nabla\left(\frac{D\theta}{Dt}\right)_{sen}\right]_H}_{(S_H)} + \underbrace{\frac{1}{\rho_0}\left[\boldsymbol{\zeta_a} \cdot \nabla\left(\frac{D\theta}{Dt}\right)_{lat}\right]}_{(L)} \tag{9}$$

where density is assumed constant ($\rho = \rho_0$) within the BL for simplicity. (7) and (8) can be substituted into (9) to give a new expression in terms of $\boldsymbol{\tau}_S$, $H_S$, and the linear function of height $S(z)$. The following depth-average operator is then applied:

$$\widehat{\frac{DP}{Dt}} = \frac{1}{h}\int\limits_0^h \left(\frac{DP}{Dt}\right) dz \tag{10}$$

With some manipulation the BL depth-averaged PV tendency equation from Cooper et al. (1992) is obtained, in the form used in Vannière et al. (2016):



$$\frac{\widehat{DP}}{Dt} = -\underbrace{\frac{\Delta\theta\hat{\mathbf{k}}\cdot(\nabla\times\boldsymbol{\tau_S})}{\rho_0{}^2 h^2}}_{(F_{EK})} - \underbrace{\frac{\boldsymbol{\tau_S}\cdot\left(\hat{\mathbf{k}}\times\nabla\theta\right)_h}{\rho_0{}^2 h^2}}_{(F_{BG})} - \underbrace{\frac{\hat{\mathbf{k}}\cdot\boldsymbol{\zeta_a}\left(z=h\right)H_S}{\rho_0 h^2}}_{(S_V)}$$

$$-\underbrace{\frac{\Delta\boldsymbol{v}\cdot\left(\hat{\mathbf{k}}\times\nabla H_S\right)}{\rho_0 h^2}}_{(S_H)} + \underbrace{\frac{1}{\rho_0 h}\int\limits_0^h \boldsymbol{\zeta_a}\cdot\nabla\dot\theta_{lat}\,dz}_{(L)} \tag{11}$$


where subscript $h$ refers to the top of the BL, $\boldsymbol{v}$ is the horizontal wind vector, and $\Delta$ refers to a change in quantity between the surface and the BL top. (11) contains 5 terms, each representing a non-conservative process which gives rise to a Lagrangian tendency of PV in the BL. Note that each term in (9) and (11) has been prescribed a shorthand name according to whether the term is associated with friction ($F$), sensible heat fluxes ($S$), or latent heat fluxes ($L$).


The $F_{EK}$ term refers to Ekman friction, capturing the impact of friction on the vertical component of vorticity. This term is proportional to the vertical Ekman pumping, and is negative for a cyclone ($\hat{\mathbf{k}}\cdot(\nabla\times\boldsymbol{\tau_S}) > 0$) with a stably stratified BL ($\Delta\theta > 0$). The $F_{BG}$ term is called baroclinic PV generation, capturing the impact of friction on the horizontal components of vorticity relating to vertical wind shear. This term is proportional to the horizontal gradient of potential temperature at the

BL top, so is large in the vicinity of fronts. The sign of this term depends on the orientation of the surface stress and the thermal wind above the BL. In mid-latitude cyclones $F_{BG}$ is positive along the warm front where the surface winds oppose the tropospheric thermal wind (e.g. Adamson et al., 2006). The $S_V$ term refers to the impact of sensible heat fluxes on the stratification in the vertical. This term is positive for a cyclone ($\zeta_a{}^z > 0$) with downward sensible heat fluxes ($H_S < 0$). Term $S_H$ is proportional to the horizontal gradient of sensible heat fluxes. Previous studies have found this term to be negligible

compared to the other terms (e.g. Plant and Belcher, 2007; Vannière et al., 2016). Term $L$ represents the effect of latent heating, which is not discussed in this paper.

## 2.6 Depth-integrated circulation budget

To understand how the BL PV tendencies impact cyclone evolution, depth-integrated circulation will be considered using control volumes centred on the cyclone. The depth-integrated circulation of a layer is equal to the volume integral of mass-

weighted PV. This is expressed mathematically in (12), where $C\Delta\theta$ is the depth-integrated circulation over a layer that is stably stratified with $\theta$-contrast $\Delta\theta$:

$$C\Delta\theta = \iiint \rho P\,dAdz \tag{12}$$

Consider an atmospheric column modelled as a cylinder centred on a cyclone, split vertically into the BL (height $h$) and free tropospheric layer above. The vector normal to the BL is $\hat{n}$, and $\hat{l}$ is the outward normal to the lateral boundary of the column.





The top of the free tropospheric layer is chosen to be an isentropic surface, $\theta_{top}$ (at height $z_{top} = z(\theta_{top})$), near the tropopause, which ensures no PV flux across it due to the impermeability theorem (Haynes and McIntyre, 1987). Here $\theta_{top} = 330$ K is used, as this level is found to reside just above the tropopause in the summer-time Arctic. There can be PV flux across the surface between the two layers (as the BL top is not necessarily an isentropic surface), and there can be PV flux across the lateral boundary. It is assumed that there are non-conservative processes in the BL, but not in the free troposphere.


Using this setup, the volume integral in (12) can be calculated following the method in Saffin et al. (2021) using Gauss' theorem and the Leibniz rule to obtain:

$$\frac{d}{dt}(C\Delta\theta)_{BL} = \iint \rho h \widehat{\frac{DP}{Dt}} \, dA - \iint_{z=h} \rho P(w - \dot{h}) \, dA - \int_0^h \oint \rho P(\boldsymbol{v} - \boldsymbol{v_b}) \cdot \hat{l} \, dldz \tag{13}$$

$$\frac{d}{dt}(C\Delta\theta)_{TROP} = \iint_{z=h} \rho P(w - \dot{h}) \, dA - \int_h^{z_{top}} \oint \rho P(\boldsymbol{v} - \boldsymbol{v_b}) \cdot \hat{l} \, dldz \tag{14}$$

where $\frac{d}{dt}(C\Delta\theta)_{BL}$ and $\frac{d}{dt}(C\Delta\theta)_{TROP}$ are the depth-integrated circulation tendencies in the BL and free tropospheric layer respectively, $w$ = vertical velocity (m s$^{-1}$), $\dot{h}$ = rate of change of BL height (m s$^{-1}$), and $\boldsymbol{v_b}$ = horizontal velocity of the lateral boundary (m s$^{-1}$). In (13) and (14) it is seen that the depth-integrated circulation of the cylinder is changed by non-conservative processes in the BL (these are neglected in the free troposphere), the vertical flux of PV across the surface between the two layers, and the horizontal fluxes of PV across the lateral boundary.

## 3 Arctic cyclone case studies

The 2020 extended summer (May–September) season is used as a sample of Arctic cyclones from which to identify case studies for further analysis. An analysis of the cyclones was performed using the ERA5 dataset (Fig. 1). Arctic cyclones are identified as vorticity maxima with 850 hPa relative vorticity, $\xi_{850}$, greater than $8 \times 10^{-5}$ s$^{-1}$ in the Arctic (> 70 °N) at least once along their track. From manual inspection the $\xi_{850}$ constraint was found to be a good filter for distinguishing synoptic-scale Arctic
cyclones from smaller-scale vorticity features.

Using this criteria, 52 Arctic cyclone tracks were identified from the 2020 summer season. The median strength in $\xi_{850}$ was $\sim 10 \times 10^{-5}$ s$^{-1}$ at maximum intensity, and the median track duration was approximately 5.4 days (Fig. 1). In Fig. 1, Arctic cyclones are also classified as low-level dominant ($Ro_T > 0$; black markers) or upper-level dominant ($Ro_T < 0$; red markers), diagnosed at the time of maximum growth rate. In 2020, 60% of the Arctic cyclones were low-level dominant (31), and 40%
were upper-level dominant (21), during development. The median track duration of the low-level dominant cyclones was 5.13 days, whilst the upper-level dominant cyclones had a longer median track duration of 6.75 days. The median strength at maxi-

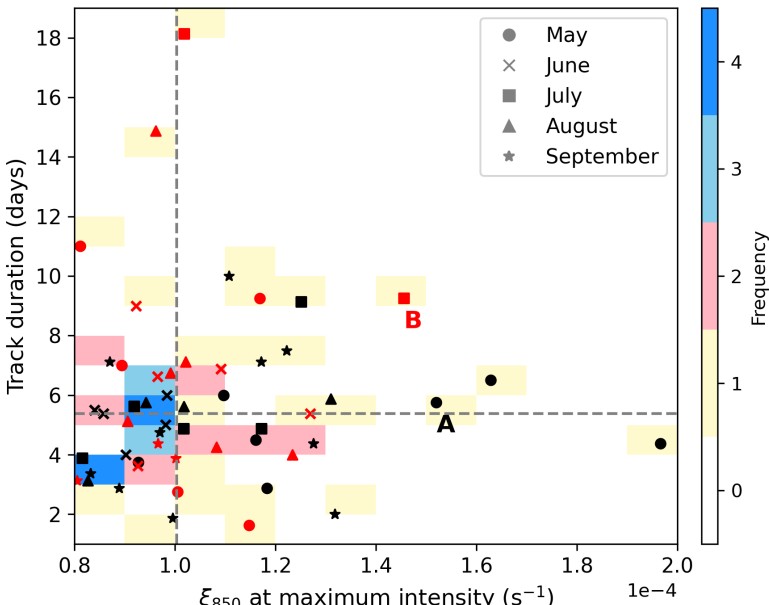

**Figure 1.** 2D histogram of $\xi_{850}$ at maximum intensity (x-axis) and track duration (y-axis) of summer 2020 Arctic cyclones from ERA5. Black and red markers refer to low-level and upper-level dominant cyclones respectively, as diagnosed by the thermal Rossby number, $Ro_T$, from the modified cyclone phase space (Section 2.4) at the time of maximum growth rate. The shading illustrates the number of cyclones that populate a region of the histogram space. The median values of $\xi_{850}$ at maximum intensity and track duration are demonstrated by the grey dashed lines. Cyclone cases A and B are annotated with the respective letter to the bottom right of the marker.

mum intensity was similar for both sets of cyclones ($\sim 10 \times 10^{-5}\,\text{s}^{-1}$).

For the purposes of this investigation two case studies are chosen, one with low-level dominant development and the other with upper-level dominant development. Cyclones A and B (annotated in Fig. 1) are selected as the strongest cyclones that spend a considerable amount of time over sea ice (note that the two strongest cyclones of the season were not chosen as case studies because the cyclone centres did not track over sea ice). Cyclone A (low-level dominant development) occurred in May 2020, and was the third strongest cyclone of summer 2020 with $\xi_{850} \sim 15 \times 10^{-5}s^{-1}$ at maximum intensity and a lifetime of

almost 6 days. Cyclone B (upper-level dominant development) was the fourth strongest cyclone with $\xi_{850} \sim 14 \times 10^{-5}s^{-1}$ at maximum intensity, with a longer track duration of almost 10 days.

The cyclone case studies were analysed in both the ERA5 reanalysis dataset and IFS forecasts (Table 1). IFS forecast start times (starting at 00Z) were selected that were closest to, but more than 24 hours before, the time of maximum growth rate

of each cyclone. Consequently, 00Z 7 May and 00Z 25 July are the chosen forecast start dates used for Cyclone A and B





**Table 1.** Summary table of Cyclones A and B tracked using filtered data from ERA5 (T5-63) and the IFS forecasts (T5-42): the time of maximum growth rate, the time of maximum intensity (as evaluated from the tracks), and the great circle separation distance between the cyclone tracks in ERA5 and the IFS forecasts at 3 selected times (forecast start date, maximum growth rate, and maximum intensity).

| **Cyclone A** | Max. growth rate | Max. intensity | Separation (km) | | |
| --- | --- | --- | --- | --- | --- |
| | | | 00Z 7 May | 12Z 8 May | 06Z 9 May |
| ERA5 | 12Z 8 May | 00Z 9 May | | | |
| IFS | 12Z 8 May | 06Z 9 May | 1290 | 40 | 63 |
| **Cyclone B** | Max. growth rate | Max. intensity | Separation (km) | | |
| | | | 00Z 25 July | 18Z 26 July | 12Z 28 July |
| ERA5 | 00Z 27 July | 00Z 28 July | | | |
| IFS | 18Z 26 July | 12Z 28 July | 929 | 96 | 60 |

respectively throughout the paper. The maximum growth rate of Cyclone A occurred at 12Z 8 May in both ERA5 and the IFS forecasts, with the system reaching maximum intensity 12 hours later at 00Z 9 May in ERA5, and 18 hours later at 06Z 9 May in the IFS forecast. In ERA5, Cyclone B underwent maximum growth at 00Z 27 July and reached maximum intensity at 00Z 28 July, compared to 18Z 26 July and 12Z 28 July respectively in the IFS forecast. In both cases the time between maximum growth rate and maximum intensity is greater in the IFS forecasts.

Cyclone A developed as part of a baroclinic wave over Western Russia before moving northwards into the Kara Sea (Fig. 2a). In both ERA5 and the IFS forecast, the cyclone develops along an elongated low-level front, however, the tracking algorithm identifies the cyclone centres at different places along the feature at 00Z 7 May (likely due to uncertainty in the position of the frontal wave along the front). As the system develops, the tracks converge. This can be seen in Table 1, with the separation reducing from 1290 km at 00Z 7 May to 40km at 12Z 8 May. This can also be seen spatially in Fig. 2a. The evolution of the cyclone structure is demonstrated by the adapted cyclone phase space (Section 2.4) in Fig. 2b. The system is initially low-level dominant (low-level warm-core) with large baroclinicity around the time of maximum growth rate. Approaching maximum intensity, and at subsequent times, the cyclone becomes more axisymmetric and ultimately becomes upper-level dominant (cold-core).

Maps of Cyclone A at 18Z 8 May 2020 (6 hours after maximum growth rate) from the IFS forecast are presented in Fig. 3. The surface cyclone is in the Kara Sea at this time, with the warm sector to the south over Russia (identified by the region of the high values of 850 hPa potential temperature to the south of the cyclone), and the warm front on the eastern flank of the warm sector (Fig. 3a). The cyclone is positioned downstream of an upper-level trough, identified by low potential temperature values on the dynamic tropopause (i.e. the 2 PVU surface; Fig. 3b). A meridional cross-section is taken through the cyclone centre from 65°N (S) to 89°N (N). In vertical cross-section (Fig. 3c) the tilted isentropes are indicative of a baroclinic zone to



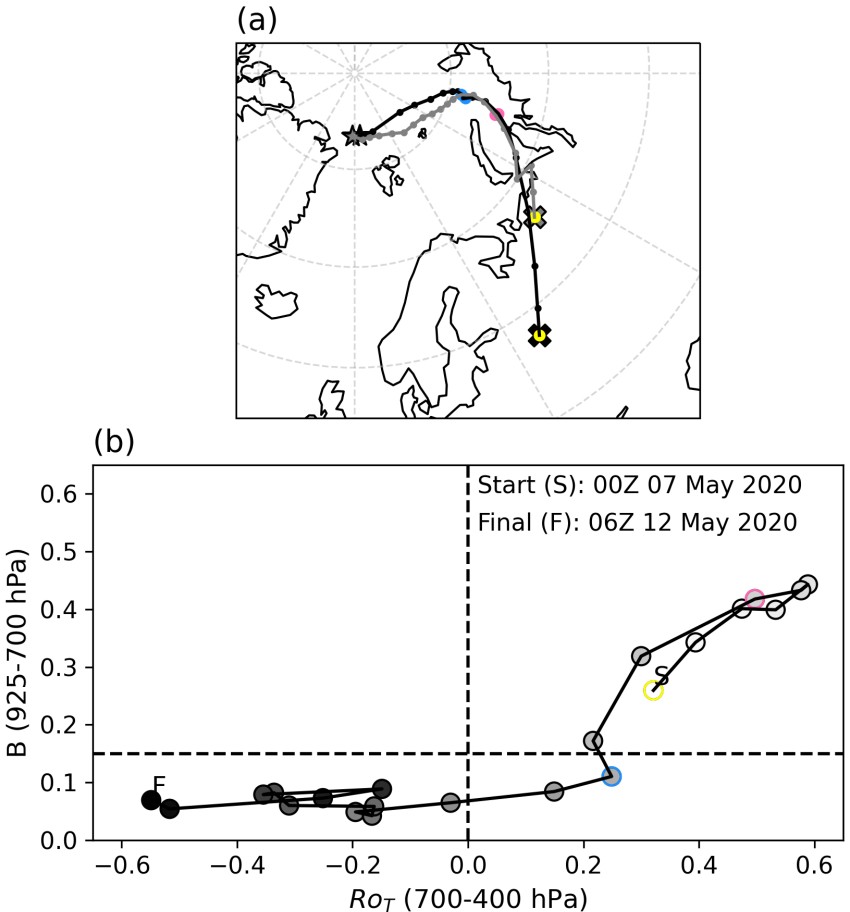

**Figure 2.** (a) 6-hourly Cyclone A tracks from ERA5 (T5-63; black line) and the IFS run starting 00Z 7 May 2020 (T5-42; grey line) over the shared temporal coverage period 00Z 7 May - 06Z 12 May 2020. The start of the tracks is marked by a cross, whilst the end is marked by a star. Note that the full length of the tracks are 18Z 6 May - 06Z 12 May (ERA5) and 00Z 7 May - 12Z 12 May (IFS). (b) Cyclone A from ERA5 in the adapted cyclone phase-space, from S (white) to F (black). The coloured points in (a) and (b) correspond to the times in Table 1: 00Z 7 May (yellow), 12Z 8 May (pink), and 06Z 9 May (blue).

the poleward side of the cyclone associated with a developing bent-back front. The upper-level trough is seen as a lowering of the tropopause to the south of the low-level cyclone (7 km height, 305 K), with the downstream ridge situated north of the low

centre (9 km height, 310 K). The dip in the isentropes over the cyclone centre (marked by the red L) indicates a warm-core structure developing, consistent with a low-level dominant cyclone and the strongest winds just above the BL. The positive PV at low levels above the low centre is reminiscent of that generated due to frictional baroclinic PV mechanism in mid-latitude cyclones (e.g. Adamson et al., 2006).



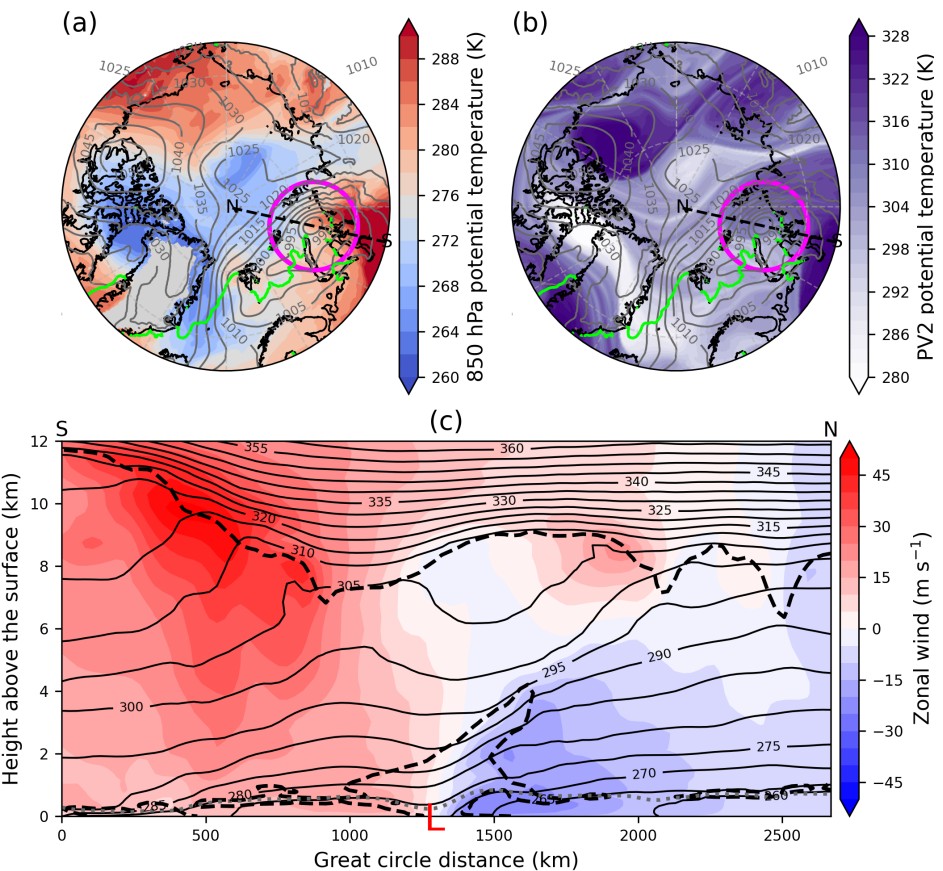

**Figure 3.** Cyclone A at 18Z 8 May 2020 from IFS run starting 00Z 7 May 2020. (a) 850 hPa potential temperature (K; shading), (b) potential temperature on the PV2 surface (K; shading), overlain with mean sea level pressure (hPa; grey contours), the sea ice edge (0.15 sea ice concentration; green contour). The magenta circle marks the 750 km radius about the cyclone centre as determined by TRACK. The black dashed lines mark the north-south cross-section taken at the longitude of the cyclone centre from S (65 °N) to N (89 °N). (c) Vertical cross-sections linearly interpolated at 100 points between S and N of zonal wind (ms$^{-1}$; shading), potential temperature (K; black solid contours), 2 PVU contour (black dashed line) and BL top (grey dotted line). Minimum mean sea level pressure is marked with a red L.

Cyclone B initially develops baroclinically north of the Arctic frontal zone along the Eurasian coastline, before interacting with a TPV in the Beaufort Sea. The cyclone tracks are presented in Fig. 4. The cyclone track from ERA5 captures the low-level baroclinic growth phase, with the vorticity maximum north of the Eurasian coastline (black line in Fig. 4a). The TPV in this case is very long-lived, and can be tracked back to 8 July 2020 (not shown). There is a secondary track from ERA5, following a low-level vorticity maximum below the pre-existing TPV (red track in Fig. 4a). This track ends once the TPV begins to interact with the low-level baroclinic disturbance at 06Z 26 July. The IFS forecast track (grey track in Fig. 4a) picks





up this vorticity maxima associated with the TPV rather than the initial baroclinic disturbance (likely due to the differences

between the reanalysis and forecast, or the different spectral filtering employed). As a result, the separation between the tracks
is initially large, but reduces as the cyclone develops (Table 1). From the ERA5 track in the cyclone phase space, Cyclone B
is initially baroclinic and low-level dominant (labelled S), but becomes upper-level dominant due to the interaction with the
TPV (Fig. 4b). After maximum intensity the cyclone obtains a long-lived (∼4 days) axisymmetric cold-core columnar vortex
structure in the Beaufort Sea. Note that the magnitude of the circularly symmetric temperature variation ($Ro_T$) is greater than
the asymmetric temperature variation ($B$; equivalently the contrast between the warm and cold sectors) in both Cyclone A and
B, i.e. the range on the x-axis is greater than that on the y-axis in Fig. 2b and 4b.

Maps of Cyclone B at 12Z 28 July 2020 (maximum intensity) from the IFS forecast are presented in Fig. 5. The surface cyclone
is located over the sea ice in the Beaufort Sea at this time, associated with a cold air mass at low-levels (Fig. 5a), and the TPV
is vertically-stacked above the cyclone (low potential temperature values in Fig. 5b). In cross-section the axisymmetric cold-
core structure is evident with isentropes bowing up throughout the troposphere centred over the cyclone (Fig. 5c). The TPV is
evident as a lowering of the tropopause to ∼4 km. The peak winds of the system are on the flanks of the TPV, consistent with
the cold-core and upper-level dominant system. The cold-core columnar vortex structure of Cyclone B looks quite different
to that of a typical mid-latitude cyclone. There is some low-level PV above the BL in the vicinity of the cyclone, but not in a
coherent region above the cyclone centre as in Cyclone A.

## 4   Results

### 4.1   Boundary layer PV tendencies

The BL PV tendencies from (11) are presented for Cyclone A at 18Z 8 May 2020 (Fig. 6). This is 6 hours after the maximum
growth rate, a period when the BL PV tendencies are large in the development of mid-latitude cyclones. The Ekman friction
term, $F_{EK}$, is negative over the cyclone centre (Fig. 6a), indicative of the Ekman pumping mechanism. The frictional baro-
clinic generation term, $F_{BG}$, is positive to the north and east of the cyclone centre, along the cyclone bent-back warm front
(Fig. 6b), as seen in the typical developing mid-latitude cyclone. The sensible heat flux term, $S_V$, is positive over the cyclone
centre and in the warm sector (Fig. 6c), where the atmosphere is warmer than the underlying surface with downward sensible
heat fluxes. Note that the $S_H$ term is much smaller than the other terms, and is therefore not shown in Fig. 6. The sum of the BL
PV tendencies is negative over the cyclone centre and to the south (behind the warm front), but is positive along the warm front
to the east and north of the cyclone (Fig. 6d). The BL PV tendencies for Cyclone A resemble those of a typical mid-latitude
cyclone (e.g. Vannière et al., 2016).


The BL PV tendencies are presented for Cyclone B at 12Z 28 July 2020, at maximum intensity after the cyclone has transi-
tioned to a vertically-stacked columnar vortex structure (Fig. 7). As in Cyclone A, the $F_{EK}$ term is negative over the cyclone
centre (Fig. 7a), as would be expected in a cyclonic weather system. However, unlike Cyclone A (and typical mid-latitude



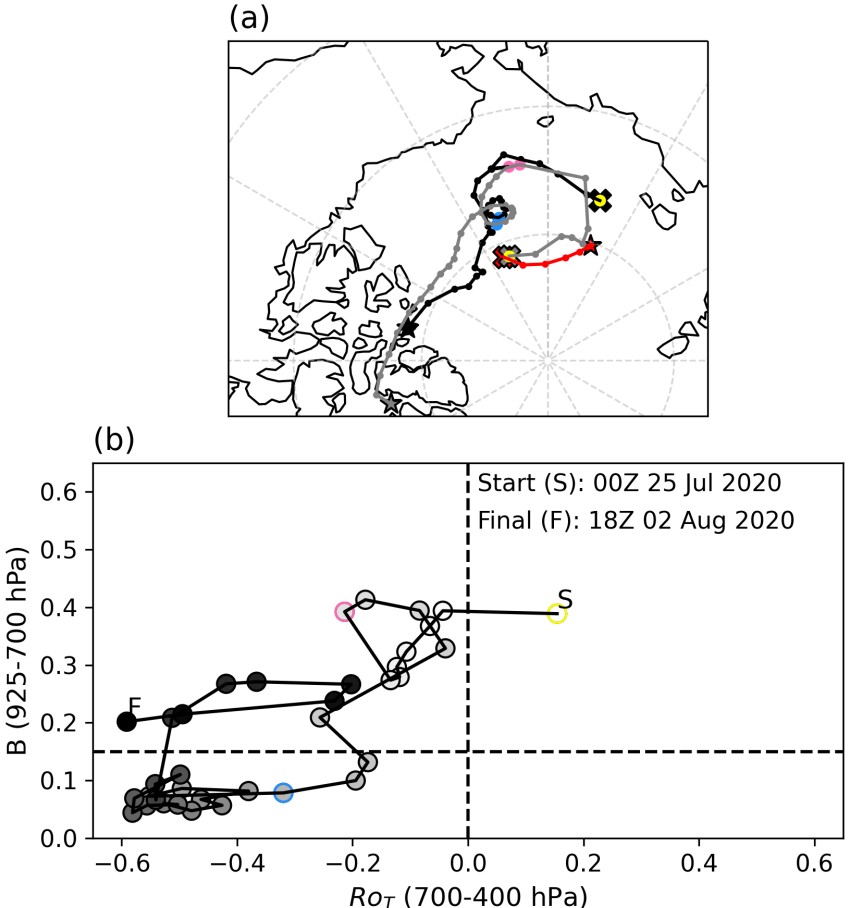

**Figure 4.** (a) 6-hourly Cyclone B tracks from ERA5 (T5-63; primary track in black line, secondary track in red line), the IFS run starting 00Z 25 July 2020 (T5-42; grey line). the shared temporal coverage period 00Z 25 July - 18Z 2 August 2020. The start of the tracks is marked by a cross, whilst the end is marked by a star. Note that the full length of the tracks are 18Z 24 July - 18Z 2 August (primary ERA5), 06Z 14 July - 06Z 26 July (secondary ERA5), and 00Z 25 July - 12Z 8 August (IFS). (b) Cyclone B from ERA5 in the adapted cyclone phase-space from S (white) to F (black). The coloured points in (a) and (b) correspond to the times in Table 1: 00Z 25 July (yellow), 18Z 26 July (pink), and 12Z 28 July (blue).

cyclones), the $F_{BG}$ term is negative, with the largest magnitude to the north and east of the cyclone in the WCB region (Fig.
7b). This is due to the vertically-stacked cold-core structure of the cyclone (Fig. 4 and 5), with the cyclonic BL winds oriented in the same direction as the cyclonic winds of the TPV directly aloft (i.e. in the same direction as the tropospheric thermal wind), in contrast to the tilted frontal structure of Cyclone A. The $S_V$ term is positive, with the largest magnitude in the WCB region (Fig. 7c), indicative of downward sensible heat fluxes over sea ice. This means that the cold air mass associated with the cyclone (Fig. 5a) is still warmer than the sea ice surface, which is locked at 0°C. The sum of the 4 BL PV tendencies is



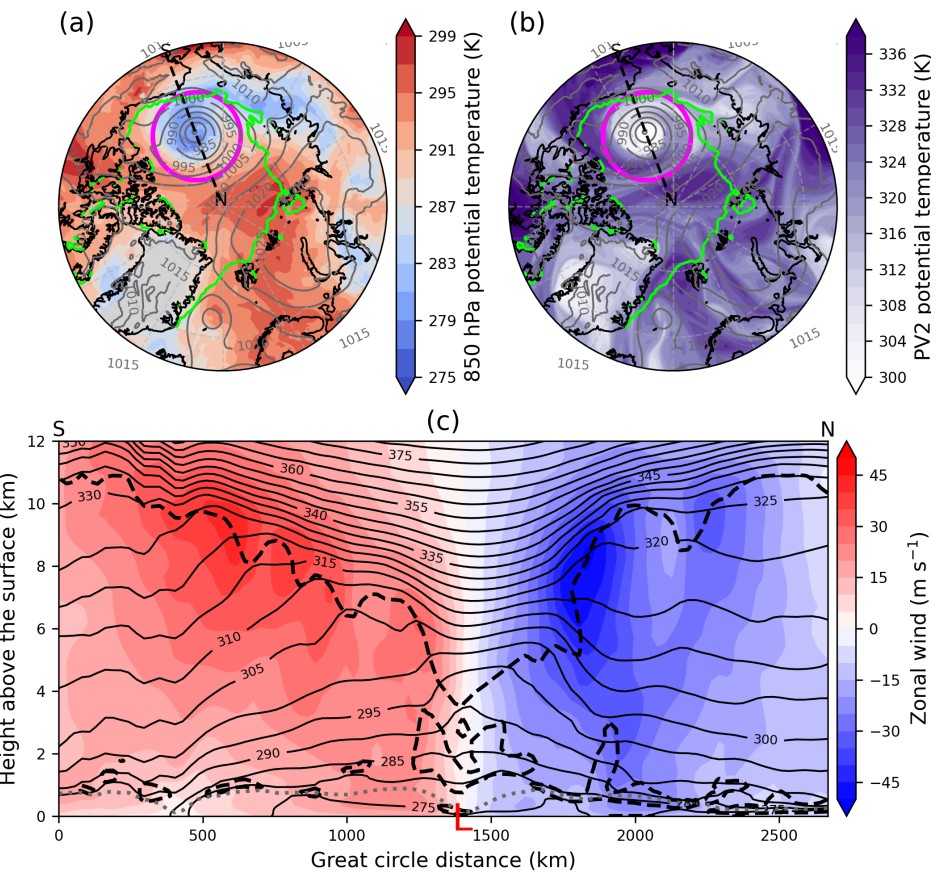

**Figure 5.** As in Fig. 3, but for Cyclone B from IFS run starting 00Z 25 July 2020, at 12Z 28 July 2020.

negative over the cyclone centre and to the north-east at this time. It is the frictional baroclinic PV generation term, $F_{BG}$ that distinguishes Cyclone B from Cyclone A (and mid-latitude cyclones).

Note that in Fig. 6 and 7 there are large PV tendencies over land far from the cyclones. This is due to greater roughness over land, and shallow BL height over orography in the model, and is not of interest in this study.

## 4.2 Cyclone depth-integrated circulation budget

A time series of the BL PV tendencies following Cyclone A throughout the system lifetime is presented in Fig. 8a, calculated as the areal mean of each term within a 750 km radius of the cyclone. Note that the BL PV tendencies have been multiplied by density and BL height to give a BL depth-integrated circulation tendency (see first term on RHS of (13)). The Ekman friction ($F_{EK}$) term is negative throughout the time series, indicative of the Ekman pumping mechanism acting throughout the

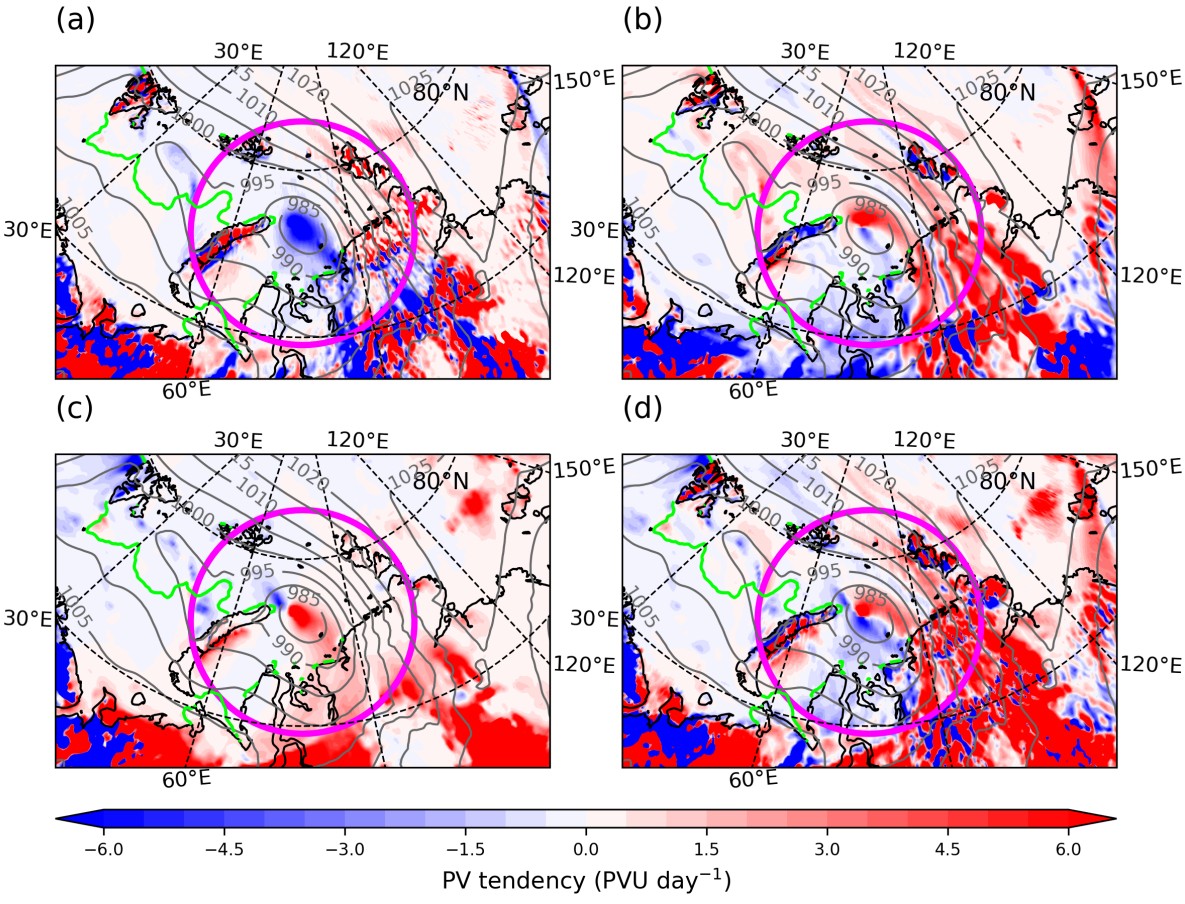

**Figure 6.** BL depth-averaged PV tendencies from Equation (11) for Cyclone A at 18Z 8 May 2020 from IFS run starting 00Z 7 May 2020. (a) $F_{EK}$, (b) $F_{BG}$, (c) $S_V$, and (d) the sum of $F_{EK}$, $F_{BG}$, $S_V$ and $S_H$ (PVU day$^{-1}$; shading), mean sea level pressure (hPa; grey contours) and the sea ice edge (0.15 sea ice concentration; green contour). The magenta circle marks 750 km from the cyclone centre.

cyclone evolution. The baroclinic PV generation ($F_{BG}$) term is large and positive during the baroclinic growth phase before the maximum intensity, with a reduced magnitude after this time (and becoming generally negative). The sensible heat flux ($S_V$) term is positive before the time of maximum intensity, dominated by the strong downward heat fluxes in the warm sector (Fig. 6c). After maximum intensity the $S_V$ term has a smaller magnitude. The $S_H$ term (proportional to the horizontal gradient of sensible heat fluxes) is also presented in Fig. 8a, and has a much smaller magnitude than the other non-conservative terms.

The sum of the BL PV tendencies is positive during the baroclinic growth phase (before maximum intensity), dominated by $F_{BG}$, and is negative once the cyclone has matured.

The depth-integrated circulation in the BL, $(C\Delta\theta)_{BL}$ (Section 2.6), of Cyclone A (Fig. 8b) increases during the baroclinic growth phase up to 6 hours before maximum intensity, and decreases after. The rate of change of $(C\Delta\theta)_{BL}$ is plotted against



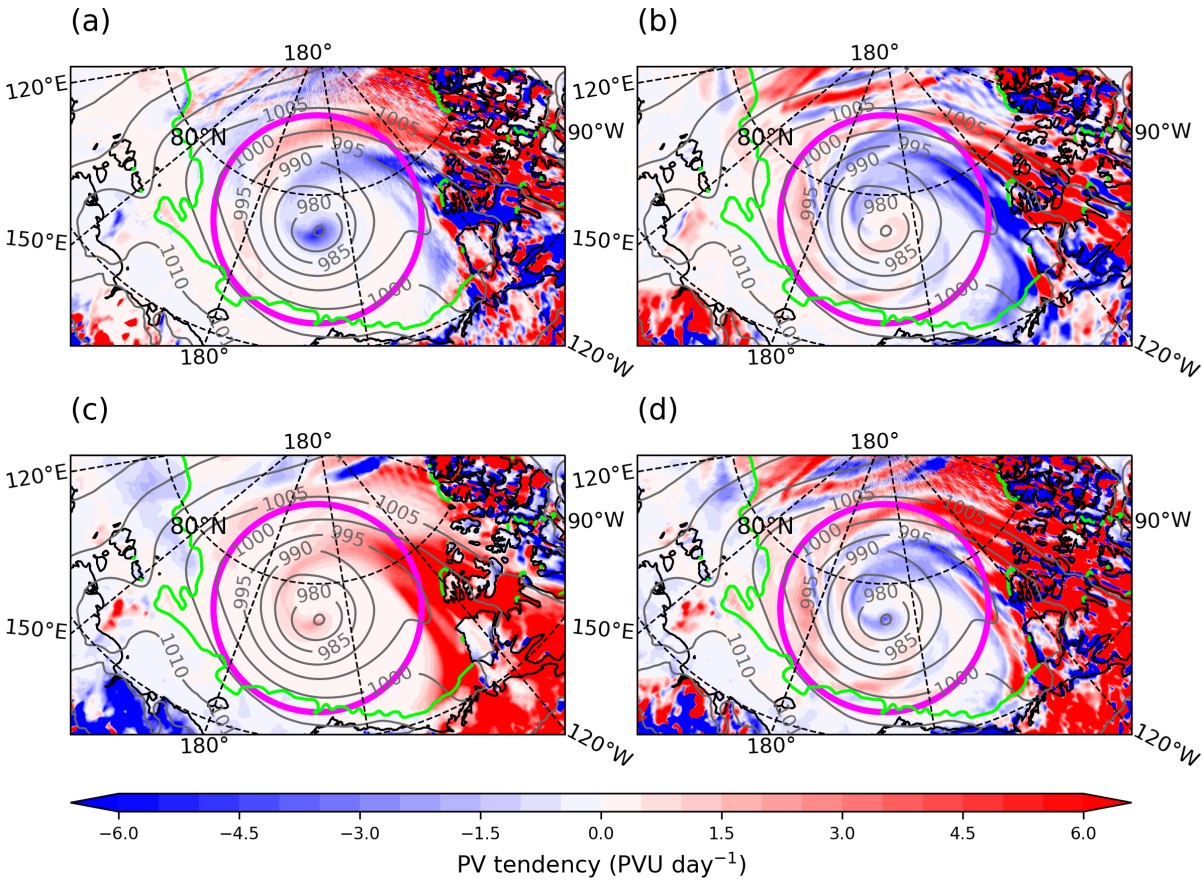

**Figure 7.** As in Fig. 6, but for Cyclone B from IFS run starting 00Z 25 July 2020 at 12Z 28 July 2020.

the BL PV tendency terms in Fig. 8a (pink dashed line), and corresponds well with the sum of the non-conservative terms (purple line). However, the sum of the non-conservative terms has a larger magnitude before maximum growth rate, which indicates that the vertical and horizontal fluxes of PV into or out of the BL control volume are large at this time, according to (13). The depth-integrated circulation in the tropospheric layer, $(C\Delta\theta)_{TROP}$, increases throughout the cyclone lifetime, even after the time of maximum intensity of the low-level cyclone. The increase in $(C\Delta\theta)_{TROP}$ is dominated by baroclinic

wave growth, which in this budget is apparent through the lateral PV fluxes into the volume. Note that $(C\Delta\theta)_{TROP}$ is approximately 15 times larger than $(C\Delta\theta)_{BL}$. The fractional rate of growth in the BL and tropospheric layer is similar up to maximum growth rate (i.e. the slope of $(C\Delta\theta)_{BL}$ and $(C\Delta\theta)_{TROP}$ are similar), which is characteristic of system growth with the BL and upper-levels coupled.

In Cyclone B (Fig. 9) the Ekman friction ($F_{EK}$) term is negative with maximum magnitude during the baroclinic growth phase at maximum growth rate (Fig. 9a). The baroclinic PV generation ($F_{BG}$) term captures two distinct periods of cyclone

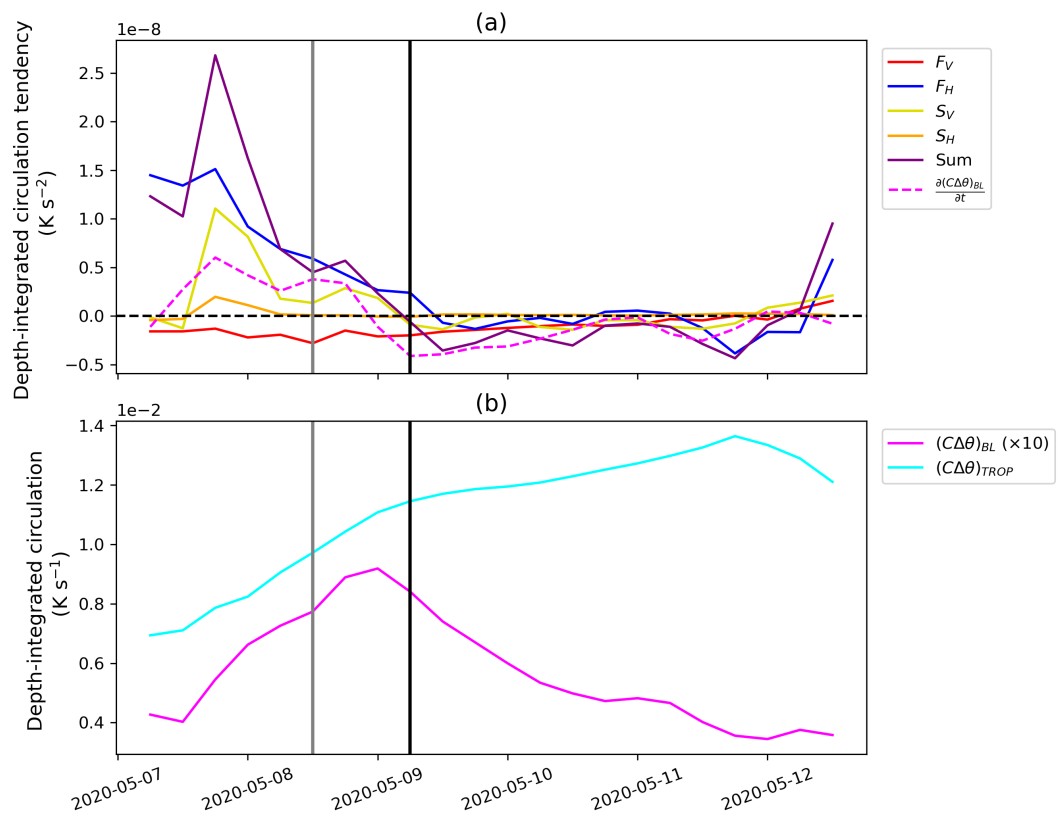

**Figure 8.** Time series of depth-integrated circulation and tendencies associated with Cyclone A, from IFS run starting 00Z 7 May 2020, from 00Z 7 May to 12Z 12 May 2020. (a) BL depth-integrated PV tendencies ($K \, s^{-2}$): $F_{EK}$ (red line), $F_{BG}$ (blue line), $S_V$ (yellow line), $S_H$ (orange line), sum (purple line) and the BL depth-integrated circulation tendency calculated explicitly (LHS of Equation 14; magenta dashed line). (b) Depth-integrated circulation ($K \, s^{-1}$) in the BL and the tropospheric layer with the 330 K isentropic surface as the top of the layer. The grey and black vertical lines correspond to the time of maximum growth rate and maximum intensity.

development. $F_{BG}$ is positive during the baroclinic growth phase, but is approximately an order of magnitude smaller than in Cyclone A (Fig. 8a), indicating reduced baroclinicity. In fact, the y-scale in Fig. 9a is almost an order of magnitude smaller than in Fig. 8a, perhaps due to weaker BL winds and surface stress in the upper-level dominant Cyclone B, resulting in smaller

PV tendencies. After the time of maximum growth rate, $F_{BG}$ reduces and rapidly becomes strongly negative, decreasing to an absolute minimum 6 hours before maximum intensity. The time series of $F_{BG}$ for Cyclone B is considerably different to that of Cyclone A at and after maximum intensity, indicating that this is an important term to understand. The sensible heat flux ($S_V$) term is largely positive due to downward sensible heat fluxes over the sea ice. The sum of the non-conservative BL





terms is positive before maximum intensity, dominated by the $S_V$ term, and is close to zero afterwards with the positive $S_V$
term reducing in magnitude and a greater (negative) contribution from $F_{BG}$. Note that the magnitude of $F_{BG}$ is greater than
$F_V$ in Cyclone A (a more baroclinic cyclone; Fig. 8a), whereas their magnitudes are comparable in Cyclone B, due to a more
barotropic structure (Fig. 9a).

Circulation in the BL, $(C\Delta\theta)_{BL}$, of Cyclone B increases during the baroclinic growth phase up to maximum intensity, and
decreases after (Fig. 9b), similar to that in Cyclone A in profile and magnitude (Fig. 8b). The rate of change of $(C\Delta\theta)_{BL}$ (Fig.
9a) corresponds well with the sum of the BL non-conservative terms. The differences between the two series is likely due to
vertical and horizontal PV flux terms, and also possibly latent heating not being considered in the budget. Unlike in Cyclone
A, $(C\Delta\theta)_{TROP}$ is relatively constant (Fig. 9b). This is related to the pre-existing TPV associated with Cyclone B. Applying
a similar reasoning used in Martínez-Alvarado et al. (2016), if the control volume containing the TPV is in an isentropic layer
(i.e. the BL top is an isentropic surface as well as the top boundary), and all the non-conservative processes lie within the circuit,
then the circulation is conserved if the lateral boundary is a material surface. When the system (TPV and low-level cyclone)
becomes a cut-off axisymmetric circuit, this condition is satisfied, and the circulation is conserved. Cyclone B largely satisfies
this condition, except during the baroclinic growth phase (i.e. the dip in $(C\Delta\theta)_{TROP}$ in Fig. 9b at maximum growth rate),
when the cyclone and TPV start to interact. This is very different to the evolution of the system circulation in Cyclone A (Fig. 8).

It is difficult to say how the BL PV tendencies contribute to the tropospheric depth-integrated circulation evolution, as the
$(C\Delta\theta)_{TROP}$ budget is dominated by baroclinic growth through the interaction between the upper and lower levels in Cyclone
A, and by the approximately conserved circulation associated with pre-existing TPV in Cyclone B. In the following sections
the impact of the BL processes on Cyclone A and B are implied by inspecting vertical cross-sections.

**4.3  Cyclone structural evolution**

Low-level (up to 700 hPa) north-south cross-sections of Cyclone A at selected times are presented in Fig. 10. The panel below
each cross-section shows the profile of the BL depth-averaged PV tendency terms (scaled as depth-integrated circulation ten-
dencies) interpolated along the section. Note that there are large PV tendencies over land (grey shading on bottom panels), but
this is not of interest in this study (as discussed in Section 4.1). At the time of maximum growth rate (12Z 8 May; Fig. 10a)
the cyclone has a baroclinic tilted structure, with tilted isentropes associated with positive PV over the cyclone centre. Winds
exceed $15 \ \mathrm{m \ s^{-1}}$ in the BL at this time. $F_{EK}$ is negative below the cyclone centre, whilst $F_{BG}$ is positive to the north (where
the BL winds are strongest). $S_V$ is positive to the south of the cyclone centre, where there are downward sensible heat fluxes
in the warm sector region. This is consistent with a region of positive PV in the BL on the southern end of the cross-section.
The sum of the BL PV tendencies is positive to the north of the cyclone, and negative to the south.

At the time of maximum intensity (06Z 9 May; Fig. 10b), the system has obtained a warm-core axisymmetric structure over
the sea ice, with isentropes dipping down over the low pressure centre. The system is very strong at low-levels this time, with

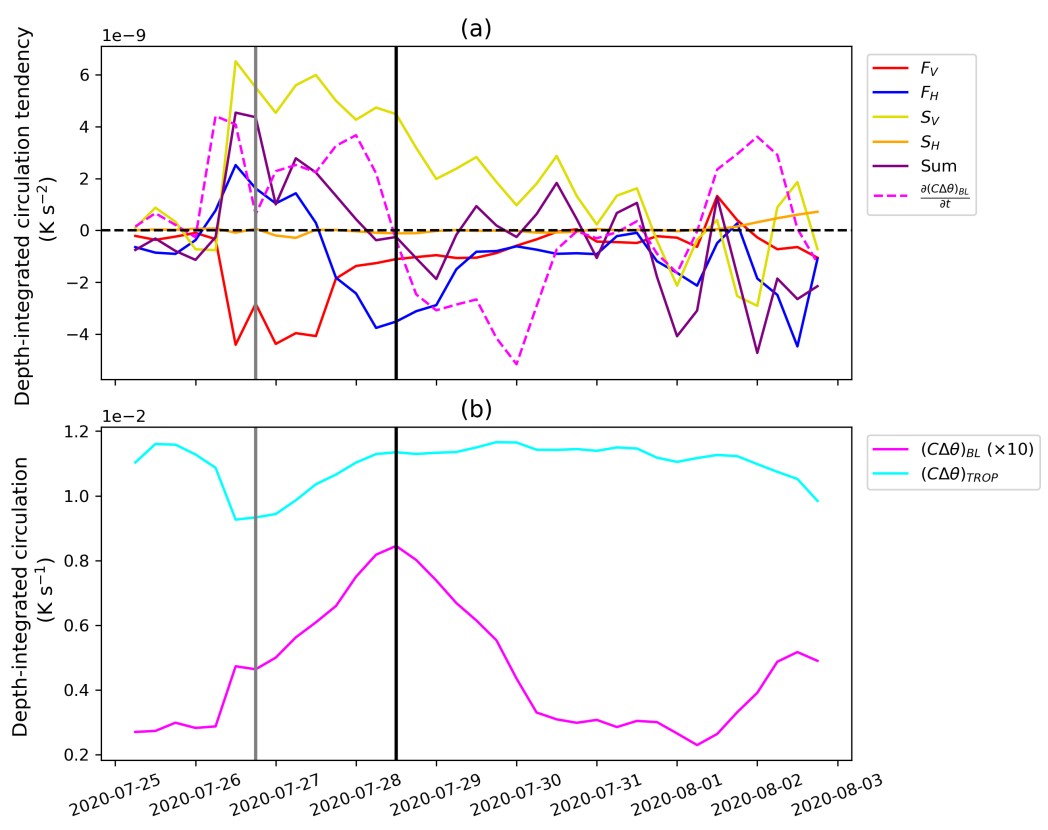

**Figure 9.** As in Fig. 8, but for Cyclone B from IFS run starting 00Z 25 July 2020, from 00Z 25 July to 00Z 2 August.

winds speeds exceeding $30\,\mathrm{m\,s^{-1}}$ in the BL. There is large positive PV above the BL constrained within a $\sim$200 km radius of the cyclone centre, associated with enhanced static stability, and is likely indicative of the frictional baroclinic PV mechanism.

As at maximum growth rate, $F_{EK}$ is negative over the cyclone centre, whilst $F_{BG}$ is positive to the north of the cyclone below the strongest BL winds. The magnitude of $S_V$ is reduced at this time. The BL height peaks where the winds are strongest, indicative of a wind-driven BL, consistent with small sensible heat fluxes. The sum of the BL PV tendencies is again positive to the north of the cyclone, consistent with positive PV there, and negative to the south of the cyclone, where there is negative PV.


24 hours after the time of maximum intensity (06Z 10 May; Fig. 10c), the cyclone has lost its warm-core, and has now developed a larger-scale cold-core structure, with the isentropes bowing upwards. This resembles the composite cold-core axisymmetric structure of summer-time Arctic cyclones after maximum intensity from Vessey et al. (2022). The isentropes have





moved upward, taking the low-level positive PV from 950 hPa up to 800 hPa. The wind field is now upper-level dominant, but
with a deep structure such that winds still exceed $20 \text{ m s}^{-1}$ in the BL. The BL PV tendencies are now reduced in magnitude
(due to weaker winds), but the $F_{BG}$ term is notably the dominant term, and is predominantly negative. The negative $F_{BG}$
term is also seen in Fig. 8 after maximum intensity, and likely reflects the cyclone's transition to an axisymmetric cold-core
structure. The cyclone retains this cold-core axisymmetric structure for 2 more days before dissipating (Fig. 2b).

This cross-section analysis is also performed for Cyclone B (Fig. 11). At the time of maximum growth rate (18Z 26 July; Fig.
11a), the cold-core structure of the pre-existing TPV is evident in the isentropes. There are baroclinic zones to the north and
south of the TPV, with tilted isentropes and positive near-surface PV. Cyclone B develops on the baroclinic zone to the south
of the TPV. The TPV is associated with a strong cyclonic wind field at the tropopause, but this does not extend to to the BL
at this time. The BL PV tendency terms are largest to the south of the section over the ocean. $F_{EK}$ is positive, indicating that
the cyclone track (as diagnosed by $\xi_{850}$) is not co-located with the minimum in mean sea level pressure at this time. $F_{BG}$ is
positive, consistent with the system undergoing baroclinic growth. With $S_V$ also being positive, there is net positive PV being
generated over the cyclone centre.

At the time of maximum intensity (12Z 28 July; Fig. 11b), the cyclone has obtained an axisymmetric cold-core structure.
The cyclonic winds about the system now extend to the lower-levels, with winds greater than $20 \text{ m s}^{-1}$ in the BL. PV is
small within the BL at this time. The magnitude of the BL PV tendency terms in Cyclone B are approximately half that of
Cyclone A, likely due to the system being upper-level dominant with weaker winds at the surface. $F_{EK}$ is negative over the
cyclone centre, and $F_{BG}$ is negative on the northern flank of the cyclone. $S_V$ is small but consistently positive over the cyclone.

72 hours after the time of maximum intensity (12Z 28 July; Fig. 11c), whilst the surface cyclone has weakened (with BL winds
of $\sim 10 \text{ m s}^{-1}$), the axisymmetric cold-core structure has amplified, with a steeper isentropic tilt on the flanks of the system.
The BL PV tendencies are small at this time, although $F_{EK}$ is notably negative over the cyclone centre, indicative of Ekman
pumping. Note that the system is not associated with a coherent accumulation of PV above the BL like in Cyclone A, and
consequently there is reduced static stability above the BL here (i.e. the isentropic surfaces are spaced further apart).

## 5   Discussion

From the results, the BL process that most obviously differs between Cyclone A and B is frictional baroclinic PV generation,
i.e. the $F_{BG}$ term. Physically, the $F_{BG}$ term is governed by the orientation of the surface winds and the low-level thermal wind.
Another form of thermal wind balance is:

$$\left( \hat{\mathbf{k}} \times \nabla \theta \right) = \frac{f_0 \theta_0}{g} \frac{\partial \boldsymbol{v}}{\partial z} \tag{15}$$





where $\frac{\partial \boldsymbol{v}}{\partial z}$ is the thermal wind vector, $\boldsymbol{v_T}$. Substituting (15) into the $F_{BG}$ term in (11) gives $F_{BG}$ explicitly in terms of the thermal wind at the top of the BL:

$$F_{BG} = \frac{f_0 \theta_0}{g \rho_0{}^2 h^2} \left( -\boldsymbol{\tau_S} \cdot \boldsymbol{v_T} \right) = -\frac{f_0 \theta_0}{g \rho_0{}^2 h^2} |\boldsymbol{\tau_S}| |\boldsymbol{v_T}| \cos \phi \qquad (16)$$

where $\phi$ is the angle between $\boldsymbol{\tau_S}$ and $\boldsymbol{v_T}$, and $\boldsymbol{\tau_S}$ is in the same direction as the surface wind ($\boldsymbol{v_S}$). Schematics of low-level dominant and upper-level dominant cyclones are presented in Fig. 12. In the low-level dominant case (Fig. 12a), in the warm

front region, the cyclonic BL wind opposes the low-level thermal wind vector just above the BL. Hence, $\boldsymbol{\tau_S}$ and $\boldsymbol{v_T}$ are opposed (i.e. 90 °$< \phi <$ 180 °). According to (16), this would yield a positive Lagrangian PV tendency ($F_{BG}$), consistent with that in Cyclone A. Now consider an axisymmetric upper-level dominant cyclone (Fig. 12b). The cyclonic BL wind is oriented in the same direction as the low-level thermal wind vector just above the BL. This means that $\boldsymbol{\tau_S}$ and $\boldsymbol{v_T}$ are oriented in the same direction (i.e. 0 °$< \phi <$ 90 °), such that the Lagrangian PV tendency ($F_{BG}$) in (16) is negative. This is consistent with the sign

of $F_{BG}$ associated with Cyclone B.

In essence, $F_{BG}$ represents changes in PV due to BL friction altering the horizontal components of vorticity. $F_{BG}$ can be written as the Lagrangian derivative of the horizontal component (considering only the $y$-component for simplicity) of (1):

$$F_{BG} = \frac{D}{Dt} \left( \frac{1}{\rho} \zeta_a{}^y \frac{\partial \theta}{\partial y} \right) = \frac{1}{\rho} \left( \frac{D(\zeta_a{}^y)}{Dt} \frac{\partial \theta}{\partial y} + \zeta_a{}^y \frac{D(\frac{\partial \theta}{\partial y})}{Dt} \right) \qquad (17)$$

where the product rule has been applied to give the RHS, and variations in density have been neglected. The horizontal vorticity in the BL, $\zeta_a{}^y$, is associated with the (zonal) vertical wind shear between the surface winds and the thermal wind at the top of the BL. Therefore $F_{BG}$ depends on the vertical wind shear across the BL ($\zeta_a{}^y$), and the horizontal temperature gradient at the BL top ($\frac{\partial \theta}{\partial y}$).

Once again, consider the warm front region of a low-level dominant cyclone, where the cyclonic BL wind opposes the low-level thermal wind vector (Fig. 12a). In this setup, $\zeta_a{}^y > 0$, and $\frac{\partial \theta}{\partial y} < 0$. Friction will act to slow down the BL winds, such that the vertical wind shear over the BL is reduced $\frac{D(\zeta_a{}^y)}{Dt} < 0$. If the system is to remain in thermal wind balance, the temperature gradient across the cyclone must also weaken: $\frac{D(\frac{\partial \theta}{\partial y})}{Dt} > 0$ (note that this yields $F_{BG} > 0$ from Equation 17). The reduction of the temperature gradient across the cyclone means that the cyclone warm-core decays.


Now consider an axisymmetric upper-level dominant, or cold-core cyclone (Fig. 12b), where the cyclonic BL wind is oriented in the same direction as the low-level (cyclonic) thermal wind vector above the BL. Here, $\zeta_a{}^y < 0$, and $\frac{\partial \theta}{\partial y} > 0$. Friction slows down the BL winds, which in this configuration, will increase the vertical wind shear over the BL: $\frac{D(\zeta_a{}^y)}{Dt} < 0$. To remain in thermal wind balance, the temperature gradient across the cyclone must increase: $\frac{D(\frac{\partial \theta}{\partial y})}{Dt} > 0$ (note that this yields $F_{BG} < 0$





from Equation 17). The increase of the temperature gradient across the cyclone means that the cyclone cold-core intensifies.

It has been shown that $F_{BG}$ has the opposite impact on cyclone thermal structure. In low-level dominant cyclones, the thermal anomaly is weakened, whereas in upper-level dominant cyclones, the thermal anomaly is amplified. The analysis demonstrates that the impact of friction depends on the cyclone structure. In both cases, $F_{BG}$ is acting to cool the thermal anomaly.


For Cyclone A, the analysis above demonstrates that $F_{BG}$ acts to decay the low-level warm-core (and amplify the cold-core anomaly once established; see negative $F_{BG}$ after maximum intensity in Fig. 8). Ekman pumping is also acting, which will also cool the system due to the rising of air and adiabatic cooling (Ekman pumping is also acting to spin-down the cyclone as it becomes equivalent barotropic). The positive $S_V$ term (before maximum intensity; Fig. 8), indicative of downward sensible

heat fluxes, also contributes to cooling with the atmosphere losing heat to the surface. For Cyclone B, $F_{BG}$ acts to amplify the cold-core, with Ekman pumping and sustained downward sensible heat fluxes over the sea ice (as indicated by negative $F_{EK}$ and positive $S_V$; Fig. 9) also contributing to low-level cooling. Hence, all of the BL PV tendencies in both cyclones are contributing to cooling the system. Low-level cooling in an axisymmetric cyclone will result in a reduction in low-level vorticity, and therefore a reduction in surface winds. This can be shown using (3), for example, in a cold-core cyclone $b' < 0$

will increase in magnitude, and therefore $\frac{\partial \xi}{\partial z} > 0$ will increase. Assuming upper-level vorticity and the layer depth stays constant, this means the low-level vorticity must decrease. Hence, the frictional processes and sensible heat fluxes are contributing to weakening the low-level cyclone after maximum intensity. Although the low-level cyclone is weakening, friction is acting to amplify a cold-core anomaly above the BL in both cyclones once they have matured. What this means for the subsequent system evolution is still an open question.


Consistent with all of the BL processes contributing to cooling the system, both cyclones obtain a vertically-stacked cold-core structure after maximum intensity (Fig. 10c and 11c) which persists for several days (Fig. 2b and 4b). This structural evolution is not seen in maturing mid-latitude cyclones. The barotropic cold-core structure after maximum intensity resembles the structural transition of summer-time Arctic cyclones in Vessey et al. (2022). Vessey et al. (2022) find that their summer-time Arctic

cyclone composite does not undergo occlusion, and suggest that summer-time Arctic cyclones may lack the dynamical forcing from the occlusion process that typically leads to the dissipation of mid-latitude cyclones. One hypothesis is that this may extend the lifetime of summer-time Arctic cyclones. This will allow BL processes (which all act to cool the thermal anomaly in Cyclones A and B) to act over a longer time period than in mid-latitude cyclones, permitting the cold-core structure to develop and persist over many days.


In Cyclone A, Ekman pumping and the baroclinic PV mechanism are both acting to increase the static stability above the BL (Boutle et al., 2015). This acts to reduce the coupling of the lower and upper levels, and eventually weaken the cyclone. In Cyclone B, the static stability above the BL is reduced compared to Cyclone A (see the large vertical spacing of the isentropes above the BL in Fig. 11c). The lower static stability would result in enhanced coupling of the lower and upper levels for longer





(compared to Cyclone A), and might explain the longer lifetime of Cyclone B (almost 10 days), compared to Cyclone A ($\sim 6$ days).

## 6    Conclusions

Previous studies have demonstrated that the evolution of mid-latitude cyclones is sensitive to BL turbulent fluxes (e.g. Valdes and Hoskins, 1988), and identified the BL processes by which the surface influences mid-latitude cyclones (e.g. Cooper et al.,

1992; Adamson et al., 2006). However, the influence of the surface and the relevant dynamical mechanisms have not yet been investigated in the context of Arctic cyclones. Differences are expected for several reasons. Firstly, surface properties (and therefore turbulent fluxes) are highly variable in the summer-time Arctic, over land, ocean, marginal ice and pack ice. Secondly, Arctic cyclones are longer lived than mid-latitude cyclones, allowing BL processes to act for longer. Thirdly, there are different cyclone growth mechanisms and morphologies in the Arctic, such that the BL processes may have different impacts

on cyclone evolution.

The purpose of this study is to characterise the BL processes acting in summer-time Arctic cyclones, and understand how they influence the structural evolution. A PV framework (derived by Cooper et al. (1992), in the Vannière et al. (2016) form) has been used, as has been used in previous studies for mid-latitude cyclones (e.g. Adamson et al., 2006; Plant and Belcher, 2007).

This PV framework in (11) reveals four boundary layer PV tendencies, each representing a BL process, associated with friction or sensible heat fluxes: $F_{EK}$ (Ekman friction), $F_{BG}$ (frictional baroclinic PV generation), $S_V$ (sensible heat fluxes), and $S_H$ (proportional the horizontal gradient sensible heat fluxes – typically smaller than the other terms). In this work, unlike previous studies, summer-time Arctic cyclones are categorised by their vorticity structure during development, as either (i) low-level dominant (low-level warm core) or (ii) upper-level dominant (tropospheric cold-core). In this study, BL processes (and their

impact on cyclone evolution) acting in two contrasting cyclone case studies from summer 2020 are investigated and compared. Cyclone A occurred in May 2020 and was low-level dominant (developing as part of a baroclinic wave off the Eurasian coastline), whilst Cyclone B occurred in July 2020 and was upper-level dominant (developing with a TPV in the Beaufort Sea). The primary tool used is the ECMWF global IFS model, focusing on a single model run for each cyclone.

The first research question (defined in Section 1) was to determine the nature of the BL processes in the different types of Arctic cyclones, and the second was to understand how these evolve with time. Both Cyclone A and B are associated with negative $F_{EK}$, and therefore Ekman pumping, which acts to spin down the cyclones throughout their lifetime (as would be expected in cyclonic weather systems). Furthermore, both cyclones are associated with positive $S_V$ due to downward sensible heat fluxes over sea ice (i.e. the atmosphere losing heat to the underlying surface), representing the generation of PV due to

increased static stability. Cyclone A is associated with positive $S_V$ before maximum intensity in the warm sector (over land and sea ice). Cyclone B is associated with positive $S_V$ over the sea ice for most of its lifetime. It is the frictional baroclinic PV generation, the $F_{BG}$ term, that differs between Cyclones A and B. The $F_{BG}$ term is positive in Cyclone A along the warm front





region during the baroclinic growth phase, where the BL winds oppose the lower tropospheric thermal wind. After maximum intensity this term reduces in magnitude, and becomes weakly negative. In Cyclone B, $F_{BG}$ is initially positive during the

baroclinic growth phase (but with a reduced magnitude than in Cyclone A, suggesting reduced baroclinicity). As the system approaches maximum intensity, the $F_{BG}$ term becomes strongly negative, as the cyclone becomes vertically stacked, with BL winds in the cyclone oriented in the same direction as the cyclonic thermal wind associated with the TPV above the BL. In both cyclones the sum of the BL PV tendencies are positive before maximum intensity, and negative afterwards.

The third research question was to compare the BL processes acting in Cyclone A and B with those in mid-latitude cyclones. The evolution of Cyclone A resembles that of mid-latitude cyclones, consistent with the same frictional processes, with negative $F_{EK}$ and positive $F_{BG}$. There is also evidence of the baroclinic PV mechanism, the dominant frictional spin-down mechanism in mid-latitude cyclones, with a region of high PV just above the BL over the low centre in Cyclone A (e.g. Adamson et al., 2006). In contrast, Cyclone B is associated with negative $F_{BG}$ due to the vertically stacked nature of the system with a TPV,

suggesting that friction is acting differently compared to Cyclone A and typical mid-latitude cyclones. Both cyclones are associated with predominantly positive $S_V$ due to downward sensible heat fluxes over the sea ice, unlike mid-latitude cyclones.

Finally, the fourth research question was to understand the impact of the BL processes on the Arctic cyclone interior evolution. It has been shown that the $F_{BG}$ term has the opposite impact on cyclone structure above the BL, depending on the cyclone

type. In Cyclone A (low-level dominant), $F_{BG}$ acts to decay the warm-core thermal anomaly, where as $F_{BG}$ acts to amplify the cold-core thermal anomaly in Cyclone B (upper-level dominant). All of the BL PV tendencies ($F_{EK}$, $F_{BG}$, $S_V$) are contributing to low-level cooling, and therefore a reduction in low-level vorticity and circulation. Although the low-level circulation of the cyclone is weakening, friction is acting to amplify a cold-core anomaly above the BL. Consistent with the BL processes contributing to cooling, both Cyclones A and B obtain a cold-core structure which persists for several days after maximum

intensity, unlike the evolution of mid-latitude cyclones. Vessey et al. (2022) have suggested that Arctic cyclones may lack the dynamical forcing to dissipate as quickly as mid-latitude cyclones, and it is hypothesised that this may allow the BL processes to act over a longer period of time. This may permit the cold-core structure to develop and persist over several days. Finally, it is hypothesised that in Cyclone B, because frictional baroclinic PV generation does not result in high PV (and high static stability) over the cyclone centre, the coupling of the lower and upper levels is prolonged, and therefore so is cyclone lifetime

(∼10 days), compared to Cyclone A (∼6 days).

This work has demonstrated that BL processes can act differently in summer-time Arctic cyclones with low-level and upper-level dominant development, and indeed differently to mid-latitude cyclones. This work is focused on two case studies. It remains to be shown in future work that the nature of the BL processes discussed here are general to low-level and upper-level

dominant development Arctic cyclones. Quantifying the relative importance of each BL process on the evolution of Arctic cyclones, and understanding the sensitivity of cyclone evolution to surface properties, are also areas for future research.





*Code and data availability.* ECMWF ERA5 reanalysis data (Hersbach et al., 2017) was retrieved via ECMWF's Meteorological Archive Retrieval System (MARS). ECMWF IFS forecast data was produced by running ECMWF's IFS model (model cycle 47r1, which was operational from 30 June 2020 to 10 May 2021). Operational forecast data is archived at ECMWF (anyone can register for access via
https://www.ecmwf.int/en/forecasts/dataset/operational-archive). The TRACK algorithm (Hodges, 1994, 1995) is available on the University of Reading's Git repository (GitLab) at https://gitlab.act.reading.ac.uk/track/track.

*Author contributions.* HLC designed the study and conducted the analysis detailed in this paper, with supervision from JM, BH, SPEK and AV. SPEK set up the ECMWF IFS model configurations which were run by HLC. HLC took responsibility to write this paper, with feedback from JM, BH, SPEK and AV.

*Competing interests.* The contact author has declared that neither they nor their co-authors have any competing interests.

*Acknowledgements.* HLC acknowledges PhD studentship funding from SCENARIO NERC Doctoral Training Partnership grant NE/S007261/1, with co-supervision and super-computing support from ECMWF. AV was funded by the Arctic Summer-time Cyclones: Dynamics and Sea-ice Interaction NERC standard grant NE/T006773/1. The authors thank Kevin Hodges for providing cyclone tracks from the TRACK algorithm for use in this study. The authors would also like to acknowledge the European Centre for Medium-Range Weather Forecasts
(ECMWF) for the production of the ERA5 reanalysis dataset.





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

**Figure 10.** North-south cross-sections of Cyclone A, from IFS run starting 00Z 7 May 2020, at the longitude of the cyclone centre from 65°N to 89°N at (a) 12Z 8 May, (b) 06Z 9 May, and (c) 06Z 10 May 2020. The top panels display potential vorticity (PVU; shading), potential temperature (K; black solid contours), zonal wind (m s$^{-1}$; blue contours), and the BL top (grey dotted line). Minimum mean sea level pressure along the section is marked with a red L. The bottom panels display the BL PV tendency terms (scaled to depth-integrated circulation tendencies) due to friction and sensible heat fluxes. The background shading denotes the surface type: land (grey), ocean (blue; sea ice concentration < 0.15), marginal ice zone (purple; sea ice concentration > 0.15 and < 0.8), and pack ice (orange; sea ice concentration > 0.8). The vertical purple line marks the cyclone centre from TRACK, and the magenta lines mark 750 km distance from the cyclone centre.



**Figure 11.** As in Fig. 11, but for Cyclone B, from IFS run starting 00Z 25 July 2020, at (a) 18Z 26 July, (b) 12Z 28 July, (c) 12Z 31 July.



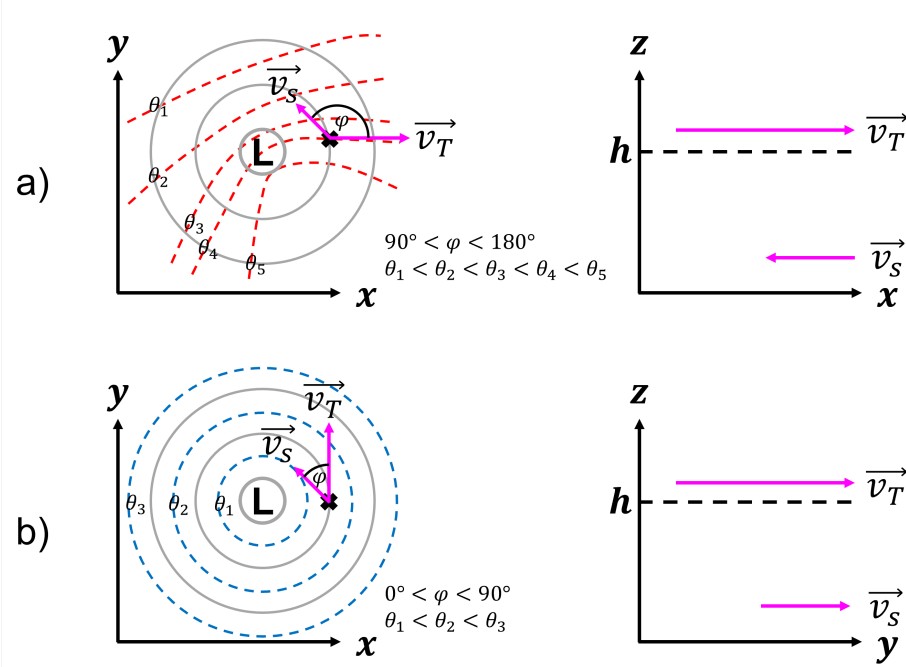

**Figure 12.** Schematics of (a) low-level dominant (warm-core) and (b) upper-level dominant (cold-core) cyclone structures. Left panels show plan views of cyclone with mean sea level pressure (grey solid contours) and potential temperature ($\theta$; dashed contours). At the point marked by the black cross, the orientation of the surface wind vector ($\boldsymbol{v_S}$; in the same direction as the surface stress, $\boldsymbol{\tau_S}$) and the thermal wind vector above the BL ($\boldsymbol{v_T}$) are demonstrated by the magenta arrows, with an angle $\phi$ between them. Right panels show vertical wind structure at this point, with the horizontal axis aligned with the largest component of $\boldsymbol{v_T}$ (in (a) this is the x-direction, in (b) this is the y-direction). The orientation of $\boldsymbol{v_S}$ and $\boldsymbol{v_T}$ are demonstrated with magenta arrows. The BL top is marked by the black dashed line (height $h$).