# Peer review of "The role of boundary layer processes in summer-time Arctic cyclones"

_Weather and Climate Dynamics, 2022_

## Referee Comment (RC1)

Review of wcd-2022-60
*'The role of boundary layer processes in summer-time Arctic cyclones"*

by Hannah L. Croad et al.

Paper in review in Weather and Climate Dynamics Discussion

**1 General Comment**

This study analyses the role of boundary layer processes for two strong Arctic cyclone case studies that track over sea ice. Both cyclone differ with respect to their structure, i.e., a warm-core and a cold-core Arctic cyclone are selected. A potential vorticity (PV) framework for PV tendencies in the boundary layer is applied, which accounts for friction (Ekmann friction and baroclinic PV generation) and sensible heat fluxes (vertical and horizontal gradients) in the boundary layer. Application of the PV framework facilitates the quantification of the effect of boundary layer processes on Arctic cyclone development. The applied framework is described well in the Methodology, and minor shortcomings of the method are discussed throughout the manuscript (e.g., PV fluxes into or out of the control volume, neglect of diabatic heating). Instantaneous PV tendencies and boundary layer averaged PV tendencies for both cyclones show that both cyclones are characterized by frictional Ekmann pumping and downward sensible heat fluxes, while frictional baroclinic PV modification differs between both cyclones and is positive for the warm-core and negative for the cold-core cyclone, respectively. In both cyclones it cools the lower troposphere, i.e. enhances the cold-core and decreases the warm-core anomaly. Although both cyclones differ with respect to their structure at maximum intensity, they both develop a persistent cold-core structure, consistent with Ekmann pumping, frictional baroclinic PV generation and downward sensible heat fluxes over sea ice, which may explain the longevity of cold-core Arctic cyclones.

The manuscript is clearly structured, comprehensive and includes well-defined research questions and the respective conclusions. It is also very well-written. The length of the manuscript is justified as it includes very comprehensive explanations of processes that act in the boundary layer. The figures could be slightly improved (see detailed comments below).

I recommend the publication of this manuscript but I have several questions that should be addressed prior to publication.

1 Introduction
The introduction is relatively long (four pages), but I find it well-written and comprehensive, and the general concepts are well explained. Thus, the length of the introduction is justified in my opinion.

2 Key questions
The introduction leads towards the clearly stated key questions. I appreciate the clear formulation of the goals of the study. Yet, in my perspective, question 3 is not a key topic addressed in this study and should be removed from the key questions. Throughout Results and Discussion, the structures from cyclones A and B are compared to what is known about mid-latitude cyclones from literature, however, the methodology is not explicitly applied to a mid-latitude cyclone in this study - which is what I would expect from question 3. Nevertheless, I find the references to mid-latitude cyclones throughout the manuscript useful.

3 Boundary layer height definition

How is the boundary layer top determined? The framework hinges on the determination of the boundary layer top. Could the authors please elaborate on how this was determined? Do the BL tops in both cyclones show a similar height, and would differences in the BL heights influence the diagnostics?

4 Presentation of figures

Generally, the figures are clearly presented. However, as the core focus lies on the comparison of cyclone A and B, I would appreciate if the authors could consistently place the corresponding figures next to each other for better comparison. I would also suggest to plot the cyclone phase-space (Figs. 2b and 4b) directly next to each other. Are cyclone phase-spaces in the IFS similar as in ERA5 (which is shown only)? Could this additionally be shown in the figure? The authors mention a lot of dates in the manuscript. Instead of referring to the date (e.g., caption Figs. 2 and 4), it would be easier for the reader if the authors would consistently refer to time of max. growth rate (pink) and max. intensity (blue). These time markers could also be potentially highlighted in a legend in the figure. Moreover, Figures 8 and 9 were better to compare if placed directly next to each other, e.g., as 2x2 panel figure. The different y-axis scaling may also be misleading at first glance. I find it difficult to identify the PV structures and respective differences in Figs. 10 and 11, and would suggest to adjust the colorbar. I appreciate the clear schematic shown in Figure 12. It may be helpful for the reader to additionally mention the respective effects on baroclinic PV modification in the figure (either in the caption or indicated in the schematic). It may also be helpful to already refer to Fig. 12 in l. 240f and l. 367.

5 Role of $F_{BG}$

The authors emphasize the different impact of $F_{BG}$ on both cyclones A and B which are characterized by a low-level dominant warm-core and upper-level dominant cold-core structure. In the maps in Figs. 6,7 the differences are present near the cyclone centers, however, I find it difficult to justify the emphasized "large" differences based on the time evolution of the cyclone-area integrated budgets, as both time series (Figs. 8 and 9) show an initial positive $F_{BG}$ tendency, and a subsequent negative $F_{BG}$ tendency - albeit with slightly different timing relative to max. intensity and differing magnitude. I wonder if the large positive values are related to processes over land (see l. 372f, and Figs. 6,7)? Were grid points over land removed from the cyclone-area integrated budgets? I wonder if this would change the evolution of PV budgets, as the values over land are substantially larger than over sea ice, which is the key focus of this study (see also next comment). Yet, I appreciate the in-depth theoretical

consideration and schematic of differences in $F_{BG}$ between cold-core and warm-core cyclones (Discussion and Fig. 12).

6 Cyclone depth-integrated circulation budget
Figure 6 shows that very large PV tendencies occur over land and coincide with the 750-km cyclone mask. Are PV tendencies over land removed from the PV tendency budget over the cyclone area show in Figs. 8-9? If a larger fraction of the cyclone mask tracks over land at certain time steps, one could imagine that the PV tendencies over land dominate the averaged PV tendencies. I wonder if the large PV tendencies for cyclone A in the early phase are related to its initial tracking over land? Are large PV tendencies over land relevant or irrelevant for the cyclone evolution if they are somewhat spatially decoupled from the cyclone center? Moreover, I find it difficult to compare the evolution of PV tendencies between both cyclones (Figs. 10 and 11). I also wonder if there is a systematic relation between the surface type and the magnitude of PV tendencies? The spatial and temporal variability in PV tendencies within each cyclone are very large, such that I find it difficult to identify the systematic differences between both systems. If the PV tendencies and cross-sections are averaged over several hours (i.e., 6-12 hours after time of max. growth rate), are some of the large-variability features smoothed out and a clearer signal emerges?

7 Discussion
The discussion summarizes the results and theoretical concepts, and explains how $F_{BG}$ acts differently in warm-core and cold-core cyclones and is associated with a cooling in both cases. I appreciate the detailed discussion which helps to put the results into context.

**2 Specific comments and technical corrections**

1. l. 89-90: Could the authors please elaborate on the concept of Ekman pumping in cyclones in the introduction, in particular the convergence at BL top and associated vortex squashing?

2. l. 216, 565: After introduction of BL as abbreviation for boundary layer, please consistently use "BL" instead of "boundary layer".

3. l. 306: "This can also be seen spatially in Fig. 2a." This sentence can be removed if Fig. 2a is referenced in the sentence before. The same applies to l. 326 "The cyclone tracks are presented in Fig. 4." .

4. l. 336ff: Could the authors please elaborate on the respective implications of the different temperature variations?

5. l. 342: Typo, please add "the" to "In cross-section".

6. l. 403: I appreciate that the authors mention the y-scale difference between Figs. 8 and 9, but I think that this difference makes it difficult to directly compare the individual tendencies (see also general comments 4 and 6).

7. l. 406f: "The time series of $F_{BG}$ for Cyclone B is considerably different to that of Cyclone A at and after maximum intensity, indicating that this is an important term to understand." From comparison of both time series, I find it difficult to conclude that $F_{BG}$ is most different, as many tendencies appear to differ substantially. Similar to the previous comment, it may help to use the same y-axis scaling for both figures. Following up on general comments 5 and 6, I wonder if there is substantial influence from interaction with land surfaces.

8. l. 463: Typo, "to to".

9. l. 464: The authors mention the larger tendencies over ocean (and previously over land), and from Figs. 10 and 11 it appears that the surface type may have an influence on the boundary layer tendencies. Do the authors find a systematic difference, and if so, how does it influence the results?

10. l. 464-465: Did the authors check if the cyclone track is not co-located with the SLP minimum?

11. l. 524: Could the authors please elaborate on the term "equivalent barotropic"?

12. Figs. 2 and 4: Please merge both tracks in one figure or combine in one large 2x2 panel figure for direct comparison of tracks A and B.

13. Figs 3a,b and 5a,b: I find the figures too small, and the colorbar for Fig. 3b is not ideally adjusted to the region of interest.

14. $F_{BG}$: In the figure legends, the $F_{BG}$ term is referred to as $F_H$, and the $F_{EK}$ term as $F_V$. I would ask the authors to make the labeling consistent.

---

## Referee Comment (RC3)

Review of wcd-2022-60

"The role of boundary layer processes in summer-time Arctic cyclones"
by
Hannah L. Croad, John Methven, Ben Harvey, Sarah P. E. Keeley, and Ambrogio
Volonté

**Recommendation: Major revisions**

**General Comments:**

The Manuscript investigates the development of Arctic cyclones and in particular the role of non-conservative processes in the boundary layer on the overall system. The authors chose an integral PV diagnostic to analyze the development as well as standard synoptic analysis. The manuscript is well written, and the figures are of good quality. While the topic is of general interest, with an increased focus on Arctic cyclones in the community, the chosen diagnostic would benefit from being put in a more physical context. Also, the specific choice of the diagnostic for the question at hand should be argued for in more detail, as other studies employed less complex analysis tools to address similar problems. Also, the potential added value of the integrated PV should be made clearer, given its rather complex nature compared to other tools, see also comments below.

One of the key findings appears to be the role of the low-level temperature structure in the cyclone development. This finding is rather interesting and appears to also explain some of the open questions about Arctic cyclones, e.g., their longevity. However, given the number of other items discussed in this manuscript as well as the complex and manifold nature of the presented material, this key finding gets a bit "lost" in the overall structure of the manuscript. Furthermore, the diagnostic tools needed to discuss this finding might not demand the complexity of the diagnostic tools chosen by the authors. It might be worthwhile to focus more on this specific aspect of the low-level temperature structure while toning down some of the other more descriptive parts addressing the development of the cyclones.

The introduction is rather long and might benefit from some rearrangements. Some aspects could be moved into other sections, for example some paragraphs in the introduction already address the methodology of the work. Related to the previous comment. If the manuscript would focus more on a specific aspect and finding, it would potentially also allow to have the introduction being more focused in that direction.

The authors should further motivate their choice of a PV framework over, e.g., an energy framework. Furthermore, investigating the impact on PV in the boundary layer is rather questionable, as the usual balance assumption breaks down in these regions that would be needed to make inferences about the implications of the changes in PV. The authors should clarify why PV should be a good and variable choice for such highly unbalanced and turbulent environments, as the main inferences about the implied development are associated to balanced PV thinking. Relatedly, in lines 426 and following, the authors state that "it is difficult to say how the BL PV tendencies contribute to the tropospheric depth-integrated circulation evolution". Given this difficulty, what is the actual benefit of this rather complex framework when trying to assess the influence of surface processes on the overall cyclone development?

The definition of the boundary layer height, upon which most of the PV-related arguments rely, is not further clarified and just given as h. How is it determined? Furthermore, the tendency of the boundary layer height is also used in the diagnostic without having clarified how this quantity is derived. The authors should specify the definition of the boundary layer height and how its tendency was calculated. Related, it would be of interest how much this boundary layer height varies within the cyclone and over the course of its lifecycle, as this has implications on the relative contributions in the PV budget.

Also, as the main interest appears to be in the horizontal circulation, i.e., the vertical component of the vorticity, it is not clear why the full three-dimensional version of the PV calculation is used. In fact, the authors also argue for the relevance of the horizontal temperature gradients, thereby including the horizontal vorticity vector component projecting onto the tilted isentropes. However, this part of the PV does not contribute to the horizontal circulation. The more tilted the isentropes, the less the PV is equivalent to the circulation one would associate with a cyclone. The authors should thus further clarify the role of these components in their argumentation. Transporting PV alone, especially if a large fraction is associated with steep isentropic slopes, cannot be directly related to changes in the horizontal circulation in a cyclone. To relate that type of PV to this kind of circulation, one would need to have significant tilting of the isentropes, to make them flatter again. It is likely that this process is involved in what is described as boundary layer ventilation in lines 116-118.

Explaining Ekman pumping in a balanced steady-state and PV-conserving framework is also a bit misleading. While the presented reasoning is self-consistent, the reduced stability in the boundary layer could have also been due to sensible heating from the surface or wind-induced mixing of the stratification. In that case, one could have attained a reduction in PV in the boundary layer without any inferences and implications about the layers aloft, i.e., no increase in stability aloft and implications for circulation, under the assumption of conserved PV. The authors should further substantiate their arguments, especially as the term Ekman pumping is used several times in the results and discussion.

Regarding the effects of surface heat fluxes, there have been recent idealized studies addressing the effects of surface fluxes on cyclone development. The authors are encouraged to put their reasoning and findings in context with these studies, which are based on both PV and energy arguments (e.g., Haualand and Spengler, 2020; Bui and Spengler, 2021). Furthermore, the authors neglect the diabatic effects of latent heating in the free troposphere. The authors should further justify this neglection, especially as latent heating often plays a significant role in cyclone development. Even though Arctic environments often feature less absolute humidity, the effects on polar cyclones can still be significant and dominant (e.g., Terpstra et al., 2015).

Regarding the depth-integrated circulation, it appears that the authors implicitly assume that the isentropes are quasi-horizontal, which contrasts with the previous emphasis of circulation projecting on rather tilted isentropic surfaces. In the extreme case of almost vertically oriented isentropes, the approximation in (12) is misleading. If the authors are mainly interested in the circulation associated with the horizontal wind components in conditions of rather flat isentropes, the previously argued relevance of the other components in the PV should be further explained and put in context. Furthermore,

density multiplied by PV is just vorticity projected onto the isentropic surface. So, the integral over an area, which would need to be level with the isentrope, would correspond to the circulation. Once isentropes feature a significant tilt, this equivalence is not exact anymore. Especially in the boundary layer, isentropes can be significantly tilted and it becomes questionable what this measure really represents. The authors should further comment on the implications of rather tilted isentropic surfaces for their diagnostic, which would become especially relevant in the boundary layer.

**Specific Comments:** (Reference to line numbers in the manuscript)

L41: Statement on previous work in that sentence needs a reference.

L61-81: This paragraph is rather long and a bit difficult to parse for the reader. Consider splitting by topics. It appears the first half focuses on TPVs, whereas the latter half makes inferences about lower and upper vorticity structures as well as warm and cold core lows.

L82-89: The paragraph first emphasizes the role of surface fluxes, but then only comments further on frictional aspects, where surface sensible and latent heat fluxes have been discussed more recently. In addition to many case studies addressing sensitivities to surface fluxes, Haualand and Spengler (2020) introduced the concept of direct and indirect effects of surface sensible and latent heat fluxes, see also Bui and Spengler (2021). It is not clear why the authors mainly emphasize the momentum fluxes.

L91: It is correct that several studies have used PV, but at least equally many have also used other measure, especially an energy framework related to the Lorenz energy cycle. The authors should provide further arguments for favoring their choice over, e.g., an energy framework.

L101: How can the authors infer that isentropes must have risen from balanced arguments? While the presented reasoning is self-consistent, the reduced stability could have also been due to sensible heating from the surface or wind-induced mixing of the stratification. In that case, one could have attained a reduction in PV in the boundary layer without any inferences and implications about the layers aloft.

L102: Why should PV be conserved in such a dynamic environment in or near the BL? And why should these layers have responded in the first place, see comment above.

L108: The association is less to horizontal temperature gradients, but to the fact that the isentropes have a significantly enough tilt so that the horizontal vorticity vector components sufficiently project onto them. Of course, there is an equivalence between the temperature gradient and the tilt of the isentropic surfaces, which will also depend on the stratification. The authors could further clarify their reasoning.

L170: How can the authors ensure that 925 hPa is above the boundary layer?

L259: How do the authors justify the assumption that there are no non-conservative processes in the free troposphere? Given that they are interested in cyclone development, diabatic processes associated to latent heat release are expected to occur in the free troposphere.

L268: See comment above about neglecting non-conservative processes in the free troposphere.

L417: The role of latent heating in the free troposphere is largely ignored in this manuscript. The authors should further justify this neglection and try to quantify it.

---

## Author Response (AR1)

**Reviewer response: The role of boundary layer processes in summer-time Arctic cyclones**

We thank the reviewers for taking the time to review our paper, and prompting what we think is a significant improvement to this manuscript. The reviewers' comments are copied below in black, with our responses in blue. Note that all line numbers and figures refer to the original manuscript that you reviewed.

**Reviewer 1:**

General comments:

This study analyses the role of boundary layer processes for two strong Arctic cyclone case studies that track over sea ice. Both cyclone differ with respect to their structure, i.e. a warm-core and a cold-core Arctic cyclone are selected. A potential vorticity (PV) framework for PV tendencies in the boundary layer is applied, which accounts for friction (Ekmann friction and baroclinic PV generation) and sensible heat fluxes (vertical and horizontal gradients) in the boundary layer. Application of the PV framework facilitates the quantification of the effect of boundary layer processes on Arctic cyclone development. The applied framework is described well in the Methodology, and minor shortcomings of the method are discussed throughout the manuscript (e.g., PV fluxes into or out of the control volume, neglect of diabatic heating). Instantaneous PV tendencies and boundary layer averaged PV tendencies for both cyclones show that both cyclones are characterized by frictional Ekmann pumping and downward sensible heat fluxes, while frictional baroclinic PV modification differs between both cyclones and is positive for the warm-core and negative for the cold-core cyclone, respectively. In both cyclones it cools the lower troposphere, i.e. enhances the cold-core and decreases the warm-core anomaly. Although both cyclones differ with respect to their structure at maximum intensity, they both develop a persistent cold-core structure, consistent with Ekmann pumping, frictional baroclinic PV generation and downward sensible heat fluxes over sea ice, which may explain the longevity of cold-core Arctic cyclones.

The manuscript is clearly structured, comprehensive and includes well-defined research questions and the respective conclusions. It is also very well-written. The length of the manuscript is justified as it includes very comprehensive explanations of processes that act in the boundary layer. The figures could be slightly improved (see detailed comments below).

I recommend the publication of this manuscript but I have several questions that should be addressed prior to publication.

1. Introduction: The introduction is relatively long (four pages), but I find it well-written and comprehensive, and the general concepts are well explained. Thus, the length of the introduction is justified in my opinion.

We thank the reviewer for their comments on the introduction. We have made some edits (in response to other reviewer comments) and have tried to be concise without cutting topics.

2. Key questions: The introduction leads towards the clearly stated key questions. I appreciate the clear formulation of the goals of the study. Yet, in my perspective, question 3 is not a key topic addressed in this study and should be removed from the key

questions. Throughout Results and Discussion, the structures from cyclones A and B are compared to what is known about mid-latitude cyclones from literature, however, the methodology is not explicitly applied to a mid-latitude cyclones in this study – which is what I would expect from question 3. Nevertheless, I find the references to mid-latitude cyclones throughout the manuscript useful.

We agree that since no calculations have been applied to mid-latitude cyclones here, it is appropriate to remove question 3 from the list of research questions (line 140 is removed). A reference to the comparison with mid-latitude cyclones is now given in the paper outline, rewriting lines 143-144 as:

"The main results are presented in Section 4, with qualitative comparison to the existing literature on mid-latitude cyclones. A more general discussion is provided in Section 5, and the study is concluded in Section 6."

As the research questions are referred to in the conclusions (Section 6), the following changes are made:

- The sentence on line 590 is rewritten as "Comparisons of the BL processes acting in Cyclone A and B with those in mid-latitude cyclones were made throughout the manuscript."
- The sentence on line 597 is rewritten as "Finally, the third research question…"

3. Boundary layer height definition: How is the boundary layer top determined? The framework hinges on the determination of the boundary layer top. Could the authors please elaborate on how this was determined? Do the BL tops in both cyclones show a similar height, and would differences in the BL heights influence the diagnostics?

BL height ($h$) is taken directly from IFS model output. A description of how the BL top is determined in the IFS is now added at the end of Section 2.2:

"In this study the BL height diagnostic from the IFS model is used, which is determined using a bulk Richardson number (ECMWF, 2020). The BL top is defined as the level at which the bulk Richardson number reaches the critical value of 0.25, i.e. the level at which the flow is no longer turbulent."

A reference to the IFS documentation has been added to the revised manuscript:

ECMWF (2020): IFS documentation Cy47r1, PART IV: PHYSICAL PROCESSES. https://www.ecmwf.int/en/elibrary/81189-ifs-documentation-cy47r1-part-iv-physical-processes.

The average BL height following each cyclone (averaged within 750 km of the cyclone centre) is presented in Figure R1. The BL height increases in both cases as the cyclone approaches maximum intensity. Furthermore, the average BL height in Cyclone A is approximately 200 m higher than in Cyclone B. This analysis demonstrates that the BL height is highest where wind speeds are the greatest (with stronger low-level winds in Cyclone A). This is indicative of stable, shear-driven BLs. The depth-integrated BL PV tendencies scale by $\frac{1}{h}$ (i.e. integral of (11) with respect to z), such that the magnitude decreases with increasing BL height. However, there are many other (interrelated) factors that control the magnitude of the terms, including the underlying land surface and cyclone strength.

A discussion of the factors controlling the magnitude of the depth-integrated BL PV terms, including BL height, is added at the beginning of Section 4.2 in the revised manuscript (please see general comment 5). Figure R1 is not included in the revised manuscript, in order to keep the paper shorter.

[Figure]

Figure R1: BL height, averaged within 750 km of the cyclone centre, from the IFS runs following (a) Cyclone A and (b) Cyclone B (red lines). The grey and black vertical lines correspond to the time of maximum growth rate and maximum intensity.

4. Presentation of figures: Generally, the figures are clearly presented. However, as the core focus lies on the comparison of cyclone A and B, I would appreciate if the authors could consistently place the corresponding figures next to each other for better comparison. I would also suggest to plot the cyclone phase-space (Figs. 2b and 4b) directly next to each other. Are cyclone phase-spaces in the IFS similar as in ERA5 (which is shown only)? Could this additionally be shown in the figure? The authors mention a lot of dates in the manuscript. Instead of referring to the date (e.g., caption Figs. 2 and 4), it would be easier for the reader if the authors would consistently refer to time of max. growth rate (pink) and max. intensity (blue). These time markers could also be potentially highlighted in a legend in the figure. Moreover, Figures 8 and 9 were better to compare if placed directly next to each other, e.g., as 2x2 panel figure. The different y-axis scaling may also be misleading at first glance. I find it difficult to identify the PV structures and respective differences in Figs. 10 and 11, and would suggest to adjust the colorbar. I appreciate the clear schematic shown in Figure 12. It may be helpful for the reader to additionally mention the respective effects on baroclinic PV modification in the figure (either in the caption or indicated in the schematic). It may also be helpful to already refer to Fig. 12 in l. 240f and l. 367.

Here we respond point by point:

(a) Placing figures next to each other: In most cases combining figures will result in the subplots being too small to see the key detail (e.g. Figures 8 and 9). The exception is Figures 2 and 4, which can be combined in one figure without significantly reducing the size of the subplots. These figures are combined, comprising Figure 2 in the revised manuscript. All other figures are not combined.

(b) Plotting IFS trajectories in cyclone phase space: In Figure R2, we plot the cyclone phase space trajectories from the IFS model instead of ERA5. The trajectory of Cyclone A in the IFS (Fig. R2b) looks similar to that from ERA5. However, the trajectory of Cyclone B

looks quite different, due to the differences in tracks of the identified features (Figure R2c). Before maximum growth rate, the IFS track has a cold-core axisymmetric structure (Figure R2d), consistent with the IFS track picking up a vorticity maximum associated with the pre-existing TPV rather than the initial low-level cyclone (see discussion in lines 325-332 of the former manuscript). This contrasts with the baroclinic structure identified on the ERA5 track. After maximum growth rate, the IFS trajectory is like that of ERA5. However, the trajectories differ again after 00Z 1 August (green markers in Figures R2c and R2d), with the IFS track picking up another vorticity feature, unlike the ERA5 track. Hence, the story from the IFS forecast trajectory is complicated by the inclusion of other features. Furthermore, the forecast likely differs somewhat from reality after 7 days. To keep the story simple (i.e. focused on Cyclone B), we have chosen to present the cyclone phase space plots from ERA5, and not the IFS (presenting both will also make the plots too cluttered).

(c) Dates in the manuscript: To make the text easier to follow, we now refer to any times in the main text as relative to maximum growth rate or maximum intensity, rather than mentioning specific dates. For the sake of completeness, dates are still given in the figure captions. We have also added legends for the coloured markers to Figure 2 in the revised manuscript (like in Figure R2), as suggested.

(d) The y-axis scaling in Figures 8 and 9: The different y-axis ranges in Figures 8 and 9 in the former manuscript are required, as the terms have a smaller magnitude in Cyclone B than for Cyclone A (i.e. it would be too difficult to see the detail of the terms for Cyclone B if the same y-axis range was used). The different y-scaling in Cyclone A and B is now discussed at the start of Section 4.2 (please see general comment 5), which will ensure that the reader is not misled. Furthermore, we add the following to the end of the caption for Figure 9 (Figure 8 in the revised manuscript): "Note that the y-scale in (a) is an order of magnitude smaller than that of Fig. 7a."

(e) Top subplot in Figures 10 and 11: To make the main PV structures more clear in Figures 10 and 11, the contour interval in the revised manuscript has been increased from 0.5 to 1 PVU. This has removed some of the finer-scale structure from the cross-sections which may distract the reader from the main PV structures.

(f) Figure 12: As suggested, we now add: "The idealised cyclone structure in (a) is associated with positive $F_{BG}$, whilst (b) is associated with negative $F_{BG}$." to the end of the caption in Figure 12 (Figure 11 in the revised manuscript). We think that Figure 12 is appropriately placed in the discussion section (Section 5). Referring to it earlier would mean having to move the figure order. Instead, we refer to Section 5 and lines 240 and 367, as suggested.

[Figure]

Figure R2: As in Figure 2 in the revised manuscript but plotted with IFS forecast model data rather than ERA5. The green markers in (c) and (d) refer to 00Z 1 August, added to supplement the response to general comment 4 part (b).

5. Role of $F_{BG}$: The authors emphasize the different impact of $F_{BG}$ on both cyclones A and B which are characterized by a low-level dominant warm-core and upper-level dominant cold-core structure. In the maps in Figs. 6,7 the differences are present near the cyclone centers, however, I find it difficult to justify the emphasized "large" differences based on the time evolution of the cyclone-area integrated budgets, as both time series (Figs. 8 and 9) show an initial positive $F_{BG}$ tendency, and a subsequent negative $F_{BG}$ tendency – albeit with slightly different timing relative to max. intensity and differing magnitude. I wonder if the large positive values are related to processes over land (see l. 372f, and Figs. 6,7)? Were grid points over land removed from the cyclone-area integrated budgets? I wonder if this would change the evolution of PV budgets, as the values over land are substantially larger than over sea ice, which is the key focus of this study (see also next comment). Yet, I appreciate the in-depth theoretical consideration and schematic of differences in $F_{BG}$ between cold-core and warm-core cyclones (Discussion and Fig. 12).

In Figures 8 and 9 in the former manuscript, grid points over land have not been removed. Figure 8 (Cyclone A) is remade here with land points not included in the calculations (Figure R3). (Note: The corresponding plot for Cyclone B is not presented here, because there are no considerable differences to discuss since the cyclone did not track over land).

The $F_{BG}$ and $S_V$ terms before maximum growth rate are reduced by an order of magnitude (Figure R3a). This confirms that the larger PV tendencies in Cyclone A do occur over land. (Note: Despite occurring over land, these PV tendencies are still relevant to the dynamics of the cyclone. Hence, Figure R3 is not presented in the revised manuscript). A discussion of this, and the factors that control the magnitude of the terms, is added at the beginning of

Section 4.2 in the revised manuscript. We also stress that as we are focused on the fundamental mechanisms in this study, it is the relative magnitudes of terms and sign of the PV tendencies in the two case studies that is the main interest, rather than the absolute magnitude (which varies with cyclone-specific details). Section 4.2 in the revised manuscript now starts as follows:

"Time series of the terms relevant to the depth-integrated PV budget of Cyclone A and B are presented in Fig. 7 and 8 respectively. The BL PV tendencies in Fig. 7a and Fig. 8a have been multiplied by density and BL height to give a BL depth-integrated PV tendency (see first term on RHS of (13)). Note that the y-scale in Fig. 7a is almost an order of magnitude larger than in Fig. 8a, due to the larger magnitudes of $F_{BG}$ and $S_V$ in Cyclone A during development. The larger magnitude of the $F_{BG}$ might be due to greater surface roughness over land, or greater baroclinicity. The surface energy balance is also changed over land, resulting in large downward sensible heat fluxes and a large $S_V$ term.

More generally, the magnitude of the BL PV tendencies is impacted by several (interrelated) factors, including the underlying surface, cyclone strength, and BL height, $h$. For instance, stronger cyclone winds correspond with larger vorticity and surface fluxes, and therefore larger PV tendencies (11). Furthermore, the depth-integrated circulation tendencies scale by $\frac{1}{h}$, such that the magnitude decreases with increasing $h$. Cyclone A has stronger low-level winds than upper-level dominant Cyclone B, but with stable shear-driven BLs also has a slightly higher $h$ (with an average value of ~800 m at maximum intensity compared to ~600 m), with opposing effects on the magnitudes of the BL PV tendencies. Clearly, the magnitude of the BL PV tendencies varies with cyclone-specific details which differ between any two cyclones. However, in this study, we are focused on the fundamental mechanisms in cyclones with contrasting structure, so it is the general evolution and sign of the BL PV tendencies that is the main interest, rather than the absolute magnitude. In the subsequent analysis, we focus on the general evolution and sign when comparing the case studies.

In Cyclone A (Fig. 7a), the Ekman friction ($F_{EK}$) term is negative throughout the time series, indicative of…"

Despite the reduction in magnitude of $F_{BG}$ in Cyclone A with the removal of grid points over land, the qualitative evolution of $F_{BG}$ is similar to that in Figure 8 in the former manuscript (i.e. in Figure R3a the $F_{BG}$ term is positive before maximum intensity, and then decreases and becomes negative afterwards). Furthermore, the main differences in the evolution of $F_{BG}$ in Cyclone A and B are unchanged, and are stated more clearly in the revised manuscript. The sentence on line 406-407 is changed to:

"The evolution of $F_{BG}$ differs to that of Cyclone A, with the transition from positive to negative occurring before maximum intensity for Cyclone B, and the negative $F_{BG}$ values having a larger magnitude (relative to the magnitude of the positive values before transition)."

Lines 373-374 are removed in the former manuscript, as the larger PV tendencies over land are now discussed at depth at the start of Section 4.2.

[Figure]

Figure R3: As in Figure 8 in the former manuscript, but with land points not included in the calculations.

6. Cyclone depth-integrated circulation budget: Figure 6 shows that very large PV tendencies occur over land and coincide with the 750-km cyclone mask. Are PV tendencies over land removed from the PV tendency budget over the cyclone area show in Figs. 8-9? If a larger fraction of the cyclone mask tracks over land at certain time steps, one could imagine that the PV tendencies over land dominate the averaged PV tendencies. I wonder if the large PV tendencies for cyclone A in the early phase are related to its initial tracking over land? Are large PV tendencies over land relevant or irrelevant for the cyclone evolution if they are somewhat spatially decoupled from the cyclone center? Moreover, I find it difficult to compare the evolution of PV tendencies between both cyclones (Figs. 10 and 11). I also wonder if there is a systematic relation between the surface type and the magnitude of PV tendencies? The spatial and temporal variability in PV tendencies within each cyclone are very large, such that I find it difficult to identify the systematic differences between both systems. If the PV tendencies and cross-sections are averaged over several hours (i.e., 6-12 hours after time of max. growth rate), are some of the large-variability features smoothed out and a clearer signal emerges?

Here we respond point by point:

(a) PV tendencies over land: The response to general comment 5 (above) demonstrates that the largest PV tendencies occur over land (during Cyclone A's development), and

that removing grid points over land from the calculations does reduce the magnitude of the averaged PV tendencies. However, despite occurring over land, these PV tendencies are still relevant to the dynamics of the cyclone (if within 750 km of the cyclone centre). For example, the baroclinic PV mechanism described by Adamson et al. (2006) demonstrates that positive PV from frictional baroclinic generation to the east of the cyclone is key for the spin-down of mid-latitude cyclones (see lines 116-119), despite being spatially decoupled from the cyclone centre. Hence, Figure R3 is not presented in the revised manuscript.

(b) Figures 10 and 11 bottom subplot: The bottom subplots in Figures 10 and 11 have been changed slightly in the former manuscript to make the PV tendencies easier to follow for the reader. A red "L", denoting the minimum sea level pressure along the section, has been added to the bottom of the subplot to help orient the reader to the centre of the cyclone (with aid from the vertical purple line denoting the cyclone centre from TRACK). Furthermore, PV tendencies outside the 750 km distance from the centre of the cyclone have been masked, as this may distract the reader from the main PV tendencies of interest (within 750 km radius of the cyclone). We think that these changes (along with the changes made in the top subplot in response to general comment 4) to Figure 10 and 11 help make the plots easier to interpret.

(c) Systematic relation between surface type and magnitude of the PV tendencies: Inspection of spatial maps indicate that the magnitudes of the friction PV tendencies are typically larger over land than over ocean or sea ice. In this study we are focused on the fundamental mechanisms, so the general evolution and sign of the PV tendencies in the two case studies is the main interest, rather than the magnitude (which varies with cyclone-specific details; see response to general comment 5).

(d) Temporal averaging of figures: Figures 8 and 9 in the former manuscript present (scaled) PV tendencies that have been averaged spatially, and present a clear signal of the evolution with time that clearly differ between Cyclone A and B (see response to general comment 5). The cross-sections in Figures 10 and 11 in the former manuscript are presented with the main purpose to demonstrate the low-level cyclone structure at the selected times (with additional instantaneous information about the PV tendencies). Temporal averaging of the cross-sections would be overly complicated given cyclone movement, and, we would argue, unnecessary, given that a clear signal of the evolution of the PV tendencies has already been established from Figures 8 and 9 in the original manuscript.

7. Discussion: The discussion summarizes the results and theoretical concepts, and explains how $F_{BG}$ acts differently in warm-core and cold-core cyclones and is associated with a cooling in both cases. I appreciate the detailed discussion which helps to put the results into context.

We thank the reviewer for their comments on the discussion.

Specific comments and technical corrections:

1. l. 89-90: Could the authors please elaborate on the concept of Ekman pumping in cyclones in the introduction, in particular the convergence at BL top and associated vortex squashing?

As some reviewer comments mention that the introduction is quite long already, in the revised manuscript we will point the reader to some sources for further information on this

topic, adding reference to Hoskins and James (2014), Section 8.7 and Holton and Hakim (2012), Section 8.4 at the end of line 89.

The following reference has been added to bibliography of revised manuscript:

Holton, J. R. and Hakim, G. J.: An Introduction to Dynamic Meteorology (5th edition), Elsevier, 2012.

2. l. 216, 565: After introduction of BL as abbreviation for boundary layer, please consistently use "BL" instead of "boundary layer".

Thank you for spotting these, we have changed "boundary layer" to "BL" on lines 216 and 565.

3. l. 306: "This can also be seen spatially in Fig. 2a." This sentence can be removed if Fig. 2a is referenced in the sentence before. The same applies to l. 326 "The cyclone tracks are presented in Fig. 4.".

Thank you for pointing these out. We have made the proposed changes:

- On line 305-6: "This can be seen in Table 1, with the separation reducing from 1290 km at the start of the forecast to 40 km at maximum growth rate, and also spatially in Fig. 2a."
- On line 325-6: "Cyclone B initially develops baroclinically north of the AFZ along the Eurasian coastline, before interacting with a TPV in the Beaufort Sea (Fig. 2c)."

4. l. 336ff: Could the authors please elaborate on the respective implications of the different temperature variations?

This statement might imply that these Arctic cyclone cases have a more circularly symmetric structure than typical baroclinic wave mid-latitude cyclones. However, this adapted cyclone phase space has not been examined across mid-latitude cyclones, so we cannot say this with any certainty. Hence, the sentence on line 336 is removed.

5. l. 342: Typo, please add "the" to "In cross-section".

We have made this correction, now written as "In the cross-section…" on line 342.

6. l. 403: I appreciate that the authors mention the y-scale difference between Figs. 8 and 9, but I think that this difference makes it difficult to directly compare the individual tendencies (see also general comments 4 and 6).

Please see the responses to general comments 4d and 5. The sentence on line 403 is removed in the revised manuscript, as this is now discussed at length at the beginning of Section 4.2 in the revised manuscript (see general comment 5).

7. l. 406f: "The times series of $F_{BG}$ for Cyclone B is considerably different to that of Cyclone A at and after maximum intensity, indicating that this is an important term to understand." From comparison of both time series, I find it difficult to conclude that $F_{BG}$ is most different, as many tendencies appear to differ substantially. Similar to the previous comment, it may help to use the same y-axis scaling for both figures. Following up on general comments 5 and 6, I wonder if there is substantial influence from interaction with land surfaces.

The difference alluded to in line 406 has now been qualified in the revised manuscript (please see general comment 5). Focusing on the evolution and sign of the PV tendencies (not the magnitude, as discussed in general comment 5), Figures 8 and 9 can be compared without changing the y-axis scaling (see response to general comment 4d).

8. l. 463: Typo, "to to".

Thanks for spotting this typo, we have removed the extra "to" on line 463.

9. l. 464: The authors mention the larger tendencies over ocean (and previously over land), and from Figs. 10 and 11 it appears that the surface type may have an influence on the boundary layer tendencies. Do the authors find a systematic difference, and if so, how does it influence the results?

There likely is a systematic relation between the surface type and magnitude of the PV tendencies (please see response to general comments 5 and 6). However, it is beyond the scope of this work to demonstrate this, and will not change the key findings of this study regarding the fundamental mechanisms.

10. l. 464-65: Did the authors check if the cyclone track is not co-located with the SLP minimum?

Yes, Figure R4 (not to be included in the manuscript) shows the map view of the $F_{EK}$ term at the time of maximum growth rate for Cyclone B. The minimum in SLP (aqua cross) is located to the south-west of the cyclone centre, as diagnosed by $\xi_{850}$ (magenta cross) at this time. The SLP minimum is associated with negative $F_{EK}$, as expected, due to Ekman pumping. As the cyclone centre is not co-located with the SLP minimum, this region of negative $F_{EK}$ does not fall along the section in Figure 11a in the former manuscript.

We rewrite the sentence on line 464-465 as: "$F_{EK}$ is positive, which is consistent with the cyclone centre (as diagnosed by $\xi_{850}$) not being co-located with the minimum in sea level pressure at this time (not shown)." in the revised manuscript.

[Figure]

Figure R4: $F_{EK}$ (PVU day$^{-1}$; shading) from IFS run starting 00Z 7 May 2020 for Cyclone B at the time of maximum growth rate. Mean sea level pressure (hPa; grey contours) and the sea ice edge (0.15 sea ice concentration; green contour) are overlaid. The magenta cross marks the cyclone centre, and the magenta circle marks 750 km from the cyclone centre. The aqua cross marks the sea level pressure minimum. The black dashed line marks the north-south cross-section taken at the longitude of the cyclone centre from S (65°N) to N (89°N).

11. l. 524: Could the authors please elaborate on the term "equivalent barotropic"?

The term "equivalent barotropic" has been used (somewhat loosely here) to refer to the vertically-stacked and axisymmetric structures of the cyclones, whilst winds still increase with height (i.e. due to the cold-core thermal wind structure).

To make this more clear to the reader, we replace "equivalent barotropic" on line 524 with "stacked in the vertical".

12. Figs. 2 and 4: Please merge both tracks in one figure or combine in one large 2x2 panel figure for direct comparison of tracks A and B.

See response to general comment 4a. We now combine Figures 2 and 4 in the former manuscript into one 2x2 figure, comprising Figure 2 in the revised manuscript.

13. Figs 3a,b and 5a,b: I find the figures too small, and the colorbar for Fig. 3b is not ideally adjusted to the region of interest.

Figures 3 and 5 in the former manuscript have been remade with subplots (a) and (b) enlarged (Figures 3 and 4 in the revised manuscript). The range of on the colour bar of Figure 3b has also been reduced to highlight more detail in the vicinity of Cyclone A.

14. $F_{BG}$: In the figure legends, the $F_{BG}$ term is referred to as $F_H$, and the $F_{EK}$ term as $F_V$. I would ask the authors to make the labelling consistent.

Thank you for spotting this (also see Reviewer 2 comment 6). This was missed when changing the notation for the finalised manuscript, with $F_V$ changed to $F_{EK}$ and $F_H$ changed to $F_{BG}$. We have made this correction in Figures 8 and 9.

**Reviewer 2:**

This paper discussed the mechanisms by which the atmospheric boundary layer affects the evolution of Arctic cyclones. This is an interesting topic, which to my knowledge has not been previously studied in a similar manner, i.e. from a potential vorticity perspective. The paper is well written and the results are interesting. I have no significant concerns, just a list of minor comments or questions which the authors may wish to consider.

Comments:

1. L111-112 - it might be worth mentioning Boutle et al. (2007, QJ) here, as that was really the first paper to discuss both the Ekman and baroclinic mechanisms acting within the same simulation (Adamson et al. mentioned both, but don't really discuss the Ekman mechanism at all, focussing on the baroclinic mechanism as the spin-down process). I suggest this because there are several places where I think the similarities/analogies to mid-latitude cyclones are slightly stronger than stated, but not all mid-latitude cyclones look like those of Adamson et al. which is the main basis for comparison. In particular:
   - L568-9 - this distinction reminds me of the Type A vs Type B classification of midlatitude systems of Petterssen & Smebye (1971), with Type A being the "true baroclinic" systems, similar to your Cyclone A and that explored by Adamson et al., whilst Type B is "upper level dominated", similar to your Cyclone B and that explored by Boutle et al. (2007 & 2015).
   - In particular, the experiment of Boutle et al. (2007) which uses a spatially uniform sea-surface temperature then becomes conceptually even more similar to your Cyclone B, where the sea-ice is providing the quasi-uniform surface temperature which limits the low level baroclinicity.
   - This is relevant for the baroclinic PV generation, where Boutle et al. (2007)'s Fig 3b (compared with your Fig 7b) shows a region of negative PV generation via this mechanism in the cyclone's warm sector. It's not explored, but I strongly suspect its creation is via the same processes you describe for Cyclone B.
   - It is also relevant for the surface heat flux PV generation, which their Fig 1b (compared with your Fig 7c) shows can provide a dominant term in the PV budget when the surface fluxes are strong (& negative) enough without much spatial variation of the surface temperature.
   - L345-346 - it does have some similarity to that presented in Fig 2a of Boutle et al. (2015), i.e. the tropopause PV structure almost reaching down to the boundary layer and interacting with the BL generated PV.

I should be clear that I don't think any of this detracts from the work that you've done, but just helps with some of the comparisons to mid-latitude systems.

Thank you for bringing these studies to our attention. We now refer to the findings from Boutle et al. (2007) in the introduction, replacing the sentence on line 111-112 as follows:

"Boutle et al. (2007) found evidence of both Ekman pumping and baroclinic PV generation in multiple model simulations, with the relative importance of each term depending on cyclone initialization. In baroclinic wave cyclones with strong low-level temperature gradients, baroclinic generation dominates (Adamson et al., 2006; Boutle et al., 2007)."

Changes are made to clarify that comparisons are made with typical baroclinic wave mid-latitude cyclones:

Lines 358-359: "The BL PV tendencies for Cyclone A resemble those of a typical baroclinic wave mid-latitude cyclone (e.g. Adamson et al., 2006; Boutle et al., 2007)."

Lines 592-593: "… the dominant frictional spin-down mechanism in baroclinic wave cyclones, …"

We also add a comparison of Cyclone B with the Boutle et al. (2007) experiment at the end of Section 4.1 (line 372), focusing on the frictional terms (to keep the story as simple as possible):

"Boutle et al. (2007) demonstrated that in a mid-latitude cyclone initialized without a meridional surface temperature gradient, the (negative) $F_{EK}$ term dominates over (mostly positive) $F_{BG}$. Cyclone B has some similarities with this experiment, with the sea ice providing a quasi-uniform surface temperature to limit low-level baroclinicity. However, unlike the Boutle et al. (2007) experiment, the vertically-stacked cold-core columnar vortex of Cyclone B results in a large region of negative $F_{BG}$ which is of a similar magnitude to that of the $F_{EK}$ term."

We also make reference to Petterssen and Smebye (1971) in the introduction, to acknowledge that our cyclone classification scheme has some similarities, but is focused purely on cyclone structure rather than mechanisms. The sentence on lines 79-81 is changed as follows:

"This is somewhat similar to the unmatched and matched classification used by Gray et al. (2021), but focuses on the vertical gradient in vorticity, rather than the identification of TPVs. These classifications are based on cyclone structure at an instant, in contrast to the Petterssen and Smebye (1971) classification of type A (low-level forcing) and type B (upper-level forcing) cyclones, which describes the development mechanisms."

The following references have been added to bibliography of revised manuscript:

Boutle, I. A., Beare, R. J., Belcher, S. E., and Plant R. S.: A note on boundary-layer friction in baroclinic cyclones, *Quart. J. Roy. Meteor. Soc.*, 133, 2137-2141, 2007.

Petterssen, S., and Smebye, S. J.: On the development of extratropical cyclones, *Quart. J. Roy. Meteor. Soc.*, 97, 457-482, 1971.

The focus of the comparison of Cyclone B with a typical mid-latitude cyclone on lines 345-346 is the cold-core columnar vortex structure (which the cyclone in Boutle et al. (2015) does not exhibit). Hence, reference to Boutle et al. (2015) is not made here, in order to keep the story as simple as possible.

2.  Section 2.5 - it's nice to present this to the reader, but it's not really new - just a repetition of what is already presented in other cited papers. You could possibly move it to an appendix to shorten the main paper and get to your own results quicker?

We agree that Section 2.5 is not novel work; the BL depth-averaged PV tendency equation was first derived by Cooper et al. (1992). However, we think that Section 2.5 is helpful for understanding the terms in the equation (i.e. emphasising the frictional vs. sensible heat flux terms and the vertical vs. horizontal components), and is complimentary for Section 2.6. Hence, we have decided to keep Section 2.5 in the main text. It is also essential to understand the schematic summarising the key result in Figure 12.

3.  L259 - is there no latent heat release happening at mid-levels within these systems? It would be good to clarify whether it is indeed small, or whether you are just ignoring it for simplicity.

Here we are ignoring latent heat release for simplicity. We agree that this should be clarified to the reader. At line 259, we now add the following: "Note that whilst there may be latent heat release happening at mid-levels within Arctic cyclones, this study focuses on the effects of the surface on cyclones through surface turbulent fluxes. Hence, latent heating is ignored here for simplicity, to isolate the impact of the dry BL processes."

4.  Table 1 - do you also only have 6 hourly data from the IFS (you state that you only have that from ERA5)? As it might be worth mentioning if so, as it probably exaggerates the differences between the two for max growth rate/intensity etc - I would hope the difference might be less if you had more frequent data, but 6 hours is the minimum difference you can 'resolve'. The statement on L300 also might not be true if you had more frequent data.

We thank the reviewer for pointing this out. The IFS model output is also 6-hourly, and this should be mentioned in Section 2.2. At line 159 we add the line: "Model runs starting at 00Z are used, with six-hourly forecasts out to 10 days.". We agree that as 6-hours is the minimum difference in time that can be resolved, the statement on line 300 might not be true. Hence, this sentence is removed in the revised manuscript.

5.  Figs 6 & 7 - not sure if it's worth pointing out somewhere that in Fig 6, the Ekman generation is almost exactly cancelled by the heat-flux generation, leaving the baroclinic generation as the only significant contributor to the total. This is another significant difference from Fig 7, where there is definitely some Ekman generation left in the residual

We adapt the discussion of the sum of the BL PV tendencies in the former manuscript to include these suggestions in the revised manuscript:

The sentence on lines 356-358 is replaced with: "The sum of the BL PV tendencies (Fig. 5d) resembles the baroclinic generation term, indicating that the Ekman generation is mostly cancelled by the sensible heat flux generation at this time."

The following is added at line 370: "Like for Cyclone A, the sum of the BL PV tendencies resembles the $F_{BG}$ term, indicating that $F_{EK}$ is mostly cancelled by $S_V$, except for the negative PV tendencies at the cyclone centre where $F_{EK}$ is the dominant term."

6. Figs 8 & 9 - the labels here don't match the text - I think you're using Fv rather than Fek and Fh rather than Fbg?

Thank you for spotting this (also see Reviewer 1 specific comment 14). This was missed when changing the notation for the finalised manuscript, with $F_V$ changed to $F_{EK}$ and $F_H$ changed to $F_{BG}$. We have made this correction in Figures 8 and 9 in the former manuscript.

7. L410 - there must be a mistake here, as you're comparing Fbg against Fv, which I think are the same - do you mean Sv or Fek?

Thank you for pointing this out. This is a typo - $F_V$ should be $F_{EK}$ (this was missed when changing the notation for the finalised manuscript, with $F_V$ changed to $F_{EK}$ and $F_H$ changed to $F_{BG}$). We have now made this change on line 411.

**Reviewer 3:**

General comments:

The Manuscript investigates the development of Arctic cyclones and in particular the role of non-conservative processes in the boundary layer on the overall system. The authors chose an integral PV diagnostic to analyze the development as well as standard synoptic analysis. The manuscript is well written, and the figures are of good quality. While the topic is of general interest, with an increased focus on Arctic cyclones in the community, the chosen diagnostic would benefit from being put in a more physical context. Also, the specific choice of the diagnostic for the question at hand should be argued for in more detail, as other studies employed less complex analysis tools to address similar problems. Also, the potential added value of the integrated PV should be made clearer, given its rather complex nature compared to other tools, see also comments below.

1. One of the key findings appears to be the role of the low-level temperature structure in the cyclone development. This finding is rather interesting and appears to also explain some of the open questions about Arctic cyclones, e.g., their longevity. However, given the number of other items discussed in this manuscript as well as the complex and manifold nature of the presented material, this key finding gets a bit "lost" in the overall structure of the manuscript. Furthermore, the diagnostic tools needed to discuss this finding might not demand the complexity of the diagnostic tools chosen by the authors. It might be worthwhile to focus more on this specific aspect of the low-level temperature structure while toning down some of the other more descriptive parts addressing the development of the cyclones.

We thank the reviewer for this comment, and we agree that the role of the low-level temperature structure is indeed a key finding of our manuscript. To emphasize this more clearly, we have re-written line 10-14 of the abstract:

"However, a third process, the frictional baroclinic generation of PV, acts differently in A and B due to differences in their low-level temperature structures. Positive PV is generated in Cyclone A near the bent-back warm front, like in typical mid-latitude cyclones. However, the same process produces negative PV tendencies in B, shown to be a consequence of the vertically-aligned axisymmetric cold-core structure."

Furthermore, whilst we agree that a range of diagnostic tools are available to understand cyclone development, we feel that the PV framework provides a unique and valuable insight here (see also response to comment 3 below). The particular formulation used here has been developed and used in the same context for examining mid-latitude cyclones (e.g. Adamson et al., 2006; Boutle et al., 2007; Plant and Belcher, 2007), allowing for an immediate and straightforward comparison of our results with that of mid-latitude cyclones.

Finally, we assume that the reviewer is referring to Section 3 as the descriptive part of the manuscript. We agree that Section 3 is descriptive in nature. However, we believe that this section is vital to (i) justify the choice of cyclone case studies, (ii) emphasise the different Arctic cyclone structures, and (iii) give a brief description of the chosen cyclones, to provide context for later analysis. Given this, and the fact that the other two reviewers did not flag this as an issue, no changes are made (beyond the response of other reviewer comments in Section 3).

2.  The introduction is rather long and might benefit from some rearrangements. Some aspects could be moved into other sections, for example some paragraphs in the introduction already address the methodology of the work. Related to the previous comment. If the manuscript would focus more on a specific aspect and finding, it would potentially also allow to have the introduction being more focused in that direction.

The paragraph on lines 61-81 has been split into two (one discusses Arctic cyclone mechanisms, the other focuses on cyclone structure) in response to specific comment 2, with some less relevant material removed. Changes have also been made in response to other reviewer comments (e.g. see reviewer 2 comment 1). Reviewer 1 (general comment 1) comments that the length of the introduction is justified, given that it is comprehensive and that the general concepts have been well explained. Hence, no other major changes have been made.

3.  The authors should further motivate their choice of a PV framework over, e.g., an energy framework. Furthermore, investigating the impact on PV in the boundary layer is rather questionable, as the usual balance assumption breaks down in these regions that would be needed to make inferences about the implications of the changes in PV. The authors should clarify why PV should be a good and variable choice for such highly unbalanced and turbulent environments, as the main inferences about the implied development are associated to balanced PV thinking. Relatedly, in lines 426 and following, the authors state that "it is difficult to say how the BL PV tendencies contribute to the tropospheric depth-integrated circulation evolution". Given this difficulty, what is the actual benefit of this rather complex framework when trying to assess the influence of surface processes on the overall cyclone development?

PV is the natural variable to discuss the dynamics of baroclinic weather systems, considering both vorticity and stratification. The central arguments in this work relate to thermal wind balance between the vorticity and potential temperature gradients, and what happens to cyclone structure when non-conservative processes modify either wind (friction) or potential temperature (diabatic processes). The benefit of using a PV framework is that structural changes within the cyclone can be inferred from the results. Changes in circulation and the constraint of thermal wind balance are not transparent in energetic frameworks and this is one of their major limitations. The following is added at line 99 to better motivate the choice of a PV framework:

"The benefit of using a PV framework is that structural changes within a cyclone can be inferred from any changes in PV, with the constraint of thermal wind balance. A PV framework is used over an energetics framework, for example, where changes in circulation and the constraint of thermal wind balance are not transparent."

The PV equation in Lagrangian form (6) is exact and no assumptions or approximations are required. Simplifying assumptions about the vertical profile of BL fluxes are made in deriving the vertically-averaged form (11) although the assumption of a linear flux gradient is not far from observed behaviour and its representation in models. The result is an equation that neatly partitions the Ekman friction term from the baroclinic term relating to horizontal potential temperature gradients. No assumptions are made about PV conservation or invertibility. We make note of this at line 234: "Also note that no assumptions are made about PV conservation or invertibility in this derivation.". Nevertheless, the volume integral of the PV equation tells us about the processes contributing to changes in circulation without needing to invoke a specific balance relation.

PV is changed in the presence of non-conservative processes. Hence, the BL PV tendencies indicate the ways in which these processes can affect circulation on larger scales simply by integrating the tendencies. The impact of the BL generated PV have been demonstrated by other modelling studies (Adamson et al., 2006; Boutle et al., 2007; Plant and Belcher, 2007). In this study no balance assumptions are needed to interpret the PV tendencies (i.e. they are just diagnostic of the BL processes).

The depth-integrated PV budget is presented to show how the BL PV tendencies are linked physically to the full PV (despite not explicitly calculating the vertical and horizontal PV flux terms). Using this, we present time evolving information about the BL PV tendencies in subplots (a) of Figures 8 and 9 (simply scaled area-averages following the cyclone). Furthermore, by considering the volume average PV ($\langle P \rangle$) in the BL and free troposphere, we get a better understanding of the impact of BL processes on the system:

- The fraction rate of growth in the BL and tropospheric layer is similar up to maximum growth rate (see line 396). These layers are coupled due to the baroclinic tilt of the system (which is continuous across both layers) – i.e. the lateral PV fluxes in (13) and (14). We show how non-conservative processes change the BL circulation, and these same processes have an indirect effect on the circulation of the tropospheric layer. To emphasise this point to the reader we now add the following at 398: "… (i.e. lateral PV fluxes in (13) and (14) are linked due to baroclinic tilt of system that is continuous across both layers). Hence, the BL processes that impact the BL circulation will have an indirect effect on the tropospheric layer also."
- After the maximum intensity, the baroclinic system tilt of the system is lost, so the coupling between the BL and tropospheric layer is reduced. The circulation becomes more axisymmetric about the cyclone (see lines 418-424). At this point, the non-conservative processes cannot have a large impact on the free tropospheric circulation (but still continue to reduce the circulation in the BL). This is emphasised to the reader at line 424: "The coupling between the BL and tropospheric layer is reduced, such that the BL processes do not significantly impact the free tropospheric circulation at this stage."

The paragraph on line 426 is intended as a transition between Section 4.2 and 4.3, and to explain that we will infer the impact of the BL processes by inspecting cyclone structure. We reword this in the revised manuscript to be more clear (and to be less 'negative' in wording):

"This analysis has revealed how the magnitude and sign of the cyclone-average BL PV tendencies evolve in time, and how the baroclinic PV generation term, $F_{BG}$, differs between the cyclones. A key question is the extent to which this difference modifies the subsequent evolution of the cyclones. Whilst a quantitative assessment of this effect, which would require a piecewise PV inversion procedure, is beyond the scope of this work, in the following section the impacts of the BL processes are qualitatively inferred by analysing the 3-dimensional structure of the cyclones and their associated PV fields. In Section 5 the $F_{BG}$ term is examined in more detail."

4. The definition of the boundary layer height, upon which most of the PV-related arguments rely, is not further clarified and just given as h. How is it determined? Furthermore, the tendency of the boundary layer height is also used in the diagnostic without having clarified how this quantity is derived. The authors should specify the definition of the boundary layer height and how its tendency was calculated. Related, it would be of interest how much this boundary layer height varies within the cyclone and over the course of its lifecycle, as this has implications on the relative contributions in the PV budget.

In this study, BL height ($h$) is taken directly from IFS model output. A description of how the BL top is determined in the IFS is now added at line 221 (please see response to reviewer 1 general comment 3). The tendency of BL height ($\dot{h}$) is involved in the formulation of the vertical PV flux across the surface between the two layers in Equations 13 and 14. However, this term (along with the horizontal PV fluxes) is not explicitly calculated (and hence, $\dot{h}$ is not calculated), and is instead discussed in general terms (e.g. see lines 391-393). This is now made clear to the reader at line 269, adding the following:

"Note that the vertical and horizontal fluxes of PV are not explicitly calculated in this work, and will be discussed in general terms as the residual in (13) and (14)."

A discussion of the BL height in each cyclone is also provided in response to reviewer 1 general comment 3 (Figure R1). The analysis indicates that the BL in both cyclones is stable, with a shear-driven BL height. This is consistent with BL height increasing as the cyclones approach maximum intensity, with stronger winds. Furthermore, this is consistent with Cyclone A (with stronger low-level winds than the upper-level dominant Cyclone B) having higher average BL heights. The impact of the BL height on the magnitude of PV tendencies is provided in the revised manuscript (please see response to reviewer 1 general comment 5). However, we stress that as we are focused on the fundamental mechanisms in this study, it is the evolution of relative magnitudes and sign of the PV tendencies in the two contrasting case studies that is the main interest, rather than the absolute magnitude (which varies with cyclone-specific details).

5. Also, as the main interest appears to be in the horizontal circulation, i.e., the vertical component of the vorticity, it is not clear why the full three-dimensional version of the PV calculation is used. In fact, the authors also argue for the relevance of the horizontal temperature gradients, thereby including the horizontal vorticity vector component projecting onto the tilted isentropes. However, this part of the PV does not contribute to

the horizontal circulation. The more tilted the isentropes, the less the PV is equivalent to the circulation one would associate with a cyclone. The authors should thus further clarify the role of these components in their argumentation. Transporting PV alone, especially if a large fraction is associated with steep isentropic slopes, cannot be directly related to changes in the horizontal circulation in a cyclone. To relate that type of PV to this kind of circulation, one would need to have significant tilting of the isentropes, to make them flatter again. It is likely that this process is involved in what is described as boundary layer ventilation in lines 116-118.

In Section 2.6, the quantity we label $C\Delta\theta$ is defined as the mass-weighted volume average of PV (Equation 12) which has been called the amount of "PV substance" by Haynes and McIntyre (1990). The budgets in Equation 13 and 14 consider the full three-dimensional PV. If the control volume is bounded above and below by isentropic surfaces, then the quantity is also the depth integral of the flow around the lateral boundary of the volume and the interpretation as Kelvin's circulation is clear (e.g., Saffin et al, 2021). Therefore, when isentropic surfaces are flat, (13) and (14) are the same as the depth-integrated circulation budget, where $C$ represents the Kelvin circulation (i.e. the azimuthal circulation around the cyclone). However, this is not a precise interpretation of C when isentropic surfaces are tilted and intersect the upper or lower boundaries of the volume, as the reviewer points out. In the revised manuscript, we now use the more general term of "depth-integrated PV budget", rather than the "depth-integrated circulation budget", to ensure that the reader is not misled. Furthermore, we change the notation in Equations 12-14 and in Figures 8 and 9, replacing $C\Delta\theta$ with $\langle P \rangle$ (where $\langle P \rangle$ is the mass-weighted volume average of PV). The opening paragraph in Section 2.6 (lines 248-251) and Equation 12 are changed as follows:

"To understand how the BL PV tendencies impact cyclone evolution, depth-integrated PV budgets will be considered using control volumes centred on the cyclone. Here we consider the quantity $\langle P \rangle$, which represents the mass-weighted volume average of PV or the "amount of PV substance" following the terminology of Haynes and McIntyre (1990):"

Furthermore the following is added at line 252, to explain how the depth-integrated PV relates to the azimuthal circulation:

"Note that $\langle P \rangle$ equals the depth-integrated circulation around the lateral boundary of the control volume only if the top and bottom boundaries of the volume are isentropic surfaces (i.e. the isentropes are flat). Since the baroclinic PV generation term depends on the gradient in potential temperature at the top of the BL (11), when this term is strong it is more precise to refer to  as PV substance than the depth-integrated circulation."

Reference added to bibliography of revised manuscript:

Haynes, P. H. and McIntyre, M.: On the conservation and impermeability theorems for potential vorticity. *J. Atmos. Sci.*, **47**, 2021-2031, 1990.

6. Explaining Ekman pumping in a balanced steady-state and PV-conserving framework is also a bit misleading. While the presented reasoning is self-consistent, the reduced stability in the boundary layer could have also been due to sensible heating from the surface or wind-induced mixing of the stratification. In that case, one could have attained a reduction in PV in the boundary layer without any inferences and implications about the layers aloft, i.e., no increase in stability aloft and implications for circulation, under the assumption of conserved PV. The authors should further substantiate their arguments,

especially as the term Ekman pumping is used several times in the results and discussion.

The explanation of Ekman pumping in the PV framework (lines 99-102) is based on a thought experiment about the impact of a frictional torque on a cyclonic vortex from Section 17.6 in Hoskins and James (2014), using adjustment to balance arguments. In this idealised setup, the impact of BL friction is isolated by assuming no other non-conservative processes are acting. Reference to this thought experiment in Hoskins and James is now made in the revised manuscript, by adapting lines 99-102 as follows:

"The impact of BL friction on a (barotropic) cyclonic vortex can be understood in the PV framework by following an idealised thought experiment from Section 17.6 in Hoskins and James (2014). Friction weakens the near-surface winds in a cyclone, reducing the azimuthal cyclone circulation (i.e. the vertical component of vorticity), and therefore reduces PV in the BL near the low centre. In a balanced state, there is both a reduction in cyclonic circulation and BL static stability. To achieve this, in the absence of other non-conservative processes, isentropes must rise, increasing the static stability above the BL."

7.  Regarding the effects of surface heat fluxes, there have been recent idealized studies addressing the effects of surface fluxes on cyclone development. The authors are encouraged to put their reasoning and findings in context with these studies, which are based on both PV and energy arguments (e.g., Haualand and Spengler, 2020; Bui and Spengler, 2021). Furthermore, the authors neglect the diabatic effects of latent heating in the free troposphere. The authors should further justify this neglection, especially as latent heating often plays a significant role in cyclone development. Even though Arctic environments often feature less absolute humidity, the effects on polar cyclones can still be significant and dominant (e.g., Terpstra et al., 2015).

The authors thank the reviewer for drawing our attention to these articles. We now refer to Haualand and Spengler (2020) in the introduction. We add the following at line 126:

"Haualand and Spengler (2020) used an idealised model to demonstrate that these negative PV tendencies act to weaken baroclinic wave development, although they found that the impact of sensible heat fluxes was relatively minor (compared to latent heating)."

We also make reference to Haualand and Spengler (2020) in the conclusion, to compare the relative importance of the $S_V$ term in Cyclone A and B. We add the following at line 596:

"The $S_V$ term is generally small in Cyclone A (except over land), consistent with the finding from Haualand and Spengler (2020) that sensible heat fluxes only have a minor impact on baroclinic wave development. In contrast, the $S_V$ term is more dominant in Cyclone B, with downward sensible heat fluxes over sea ice over the entire cyclone."

Reference added to manuscript:

Haualand, K. F., and Spengler, T.: Direct and indirect effects of surface fluxes on moist baroclinic development in an idealized framework. *J. Atmos. Sci.*, **77**, 3211-3225, 2020.

We do not refer to Bui and Spengler (2021), as the study uses an energetics framework to diagnose cyclone evolution, and so is not relevant to this work.

Regarding the neglection of latent heat fluxes in this study, please see response to reviewer 2 comment 3. Whilst there may be latent heat release happening in the free troposphere, our study focuses on the effects of the surface on Arctic cyclones, including surface heat fluxes and frictional processes. Non-conservative processes in the free troposphere (including latent heating) are not examined and impact of the dry BL processes is isolated. This is now clarified to the reader at line 259, 268 and 417 (see response to specific comments 9, 10 and 11).

8. Regarding the depth-integrated circulation, it appears that the authors implicitly assume that the isentropes are quasi-horizontal, which contrasts with the previous emphasis of circulation projecting on rather tilted isentropic surfaces. In the extreme case of almost vertically oriented isentropes, the approximation in (12) is misleading. If the authors are mainly interested in the circulation associated with the horizontal wind components in conditions of rather flat isentropes, the previously argued relevance of the other components in the PV should be further explained and put in context. Furthermore, density multiplied by PV is just vorticity projected onto the isentropic surface. So, the integral over an area, which would need to be level with the isentrope, would correspond to the circulation. Once isentropes feature a significant tilt, this equivalence is not exact anymore. Especially in the boundary layer, isentropes can be significantly tilted and it becomes questionable what this measure really represents. The authors should further comment on the implications of rather tilted isentropic surfaces for their diagnostic, which would become especially relevant in the boundary layer.

Please see general comment 5. Equation 12 and the budgets in Equation 13 and 14 consider the volume integral of the full PV (and C is not the circulation around the lateral boundary of the volume except in the special case that the volume is an isentropic layer) – no assumptions have been made about the slope of the isentropic surfaces. To ensure the reader is not misled, in the revised manuscript we now use the more general term of "depth-integrated PV budget", rather than the "depth-integrated circulation budget". Furthermore, we change the notation in Equations 12-14 and in Figures 8 and 9, replacing $C\Delta\theta$ with $\langle P \rangle$ (where $\langle P \rangle$ is the mass-weighted volume average of PV).

Specific comments:

1. L41: Statement on previous work in that sentence needs a reference.

This sentence is leading on to the next few sentences regarding the work by Yamagami et al. (2018a), Capute and Torn (2021), and Yamagami et al. (2018b). To make this more obvious to the reader, we add "For instance," to the start of the sentence on line 42.

2. L61-81: This paragraph is rather long and a bit difficult to parse for the reader. Consider splitting by topics. It appears the first half focuses on TPVs, whereas the latter half makes inferences about lower and upper vorticity structures as well as warm and cold core lows.

In the revised manuscript this paragraph has been split into two as suggested, with the first part focusing on cyclone mechanisms (in particular TPVs), and the second focusing on the vorticity structure of Arctic cyclones. Furthermore, some detail has been removed (e.g. the

details about the AFZ) in an attempt to shorten this section and make the story easier to follow for the reader:

"Many of the growth mechanisms of summer-time Arctic cyclones are the same for mid-latitude cyclones, such as baroclinic instability and lee cyclogenesis. However, sustained cyclone interaction with tropopause polar vortices (TPVs), long-lived vortices on the tropopause with horizontal scales of less than 1500 km (Cavallo and Hakim, 2009), is a characteristic of the Arctic (where there is typically an absence of a strong zonal jet stream in the upper troposphere). Gray et al. (2021) classified Arctic cyclones as being either (i) "unmatched" or (ii) "matched" with a TPV during development, using a statistical matching criterion based on a threshold distance between tracked TPVs and low-level cyclones. It was found that unmatched cyclones are initially dominated by low-level vorticity, such that vorticity decreases with height. These cyclones occur most commonly along the northern coast of Eurasia (Fig. 7 in Gray et al. (2021)), in association with high baroclinicity on the Arctic Frontal Zone (AFZ; Serreze et al., 2001; Day and Hodges, 2018). In contrast, matched cyclones are dominated by upper-level vorticity (vorticity increases with height). Matched cyclones are associated with reduced tilt and baroclinicity, and a single columnar vortex structure at maximum intensity (like the summer-time Arctic cyclone composite in Vessey et al. (2022)). Matched cyclones track most frequently along the North American coastline (Fig. 7 in Gray et al. (2021)), consistent with the climatological location of TPVs (Cavallo and Hakim, 2010).

In this study, Arctic cyclones will be classified in terms of their vorticity structure during development, as either (i) low-level dominant or (ii) upper-level dominant. Note that by thermal wind balance, these cyclones have (i) low-level warm cores (i.e. a horizontal temperature maximum) and (ii) tropospheric cold cores respectively. This is somewhat similar to the unmatched and matched classification used by Gray et al. (2021), but focuses on the vertical gradient in vorticity, rather than the identification of TPVs. These classifications are based on cyclone structure at an instant, in contrast to the Petterssen and Smebye (1971) classification of type A (low-level forcing) and type B (upper-level forcing) cyclones, which describes the development mechanisms."

3. L82-89: The paragraph first emphasizes the role of surface fluxes, but then only comments further on frictional aspects, where surface sensible and latent heat fluxes have been discussed more recently. In addition to many case studies addressing sensitivities to surface fluxes, Haualand and Spengler (2020) introduced the concept of direct and indirect effects of surface sensible and latent heat fluxes, see also Bui and Spengler (2021). It is not clear why the authors mainly emphasize the momentum fluxes.

This paragraph begins by stating the importance of turbulent fluxes generally in cyclones. The paragraph then goes on to discuss the turbulent momentum fluxes (i.e. friction) in particular. We believe the decision to discuss the frictional processes first is justified, because these are most relevant to the results of this paper (i.e. the key finding is that the frictional baroclinic generation differs between Cyclones A and B). Sensible heat fluxes are discussed later, in the paragraph starting at line 124. We make this obvious to the reader by adding the following at line 85:

"Here we discuss the momentum fluxes first (sensible heat fluxes are discussed in a later paragraph)."

4. L91: It is correct that several studies have used PV, but at least equally many have also used other measure, especially an energy framework related to the Lorenz energy cycle. The authors should provide further arguments for favoring their choice over, e.g., an energy framework.

Please see response to general comment 3.

5. L101: How can the authors infer that isentropes must have risen from balanced arguments? While the presented reasoning is self-consistent, the reduced stability could have also been due to sensible heating from the surface or wind-induced mixing of the stratification. In that case, one could have attained a reduction in PV in the boundary layer without any inferences and implications about the layers aloft.

Please see general comment 6.

6. L102: Why should PV be conserved in such a dynamic environment in or near the BL? And why should these layers have responded in the first place, see comment above.

Please see general comment 6.

7. L108: The association is less to horizontal temperature gradients, but to the fact that the isentropes have a significantly enough tilt so that the horizontal vorticity vector components sufficiently project onto them. Of course, there is an equivalence between the temperature gradient and the tilt of the isentropic surfaces, which will also depend on the stratification. The authors could further clarify their reasoning.

In thermal wind balance the horizontal vorticity vector is proportional to the horizontal potential temperature gradient (so two ways of saying the same thing). We adapt line 108 to make mention of tilted isentropic surfaces: "This process is called "frictional baroclinic PV generation" and is most prominent in regions of strong horizontal temperature gradients where isentropes have significant tilt, such as fronts."

8. L170: How can the authors ensure that 925 hPa is above the boundary layer?

It is assumed that 925 hPa is above the BL (this seems reasonable given the cross-sections Figs. 10 and 11 in the former manuscript), although this cannot be guaranteed. We adapt line 170 in the revised manuscript to clarify this to the reader: "(assumed to be above the BL but below the 'steering' level)".

Furthermore, we adapt lines 187-188 in the former manuscript to "(assumed to be above the 'steering' level but below the tropopause)".

9. L259: How do the authors justify the assumption that there are no non-conservative processes in the free troposphere? Given that they are interested in cyclone development, diabatic processes associated to latent heat release are expected to occur in the free troposphere.

Whilst there might be latent heat release happening in the free troposphere, this study focuses on the effects of the surface on Arctic cyclones. Hence latent heat release is not examined for simplicity, so the impact of the dry BL processes is isolated. Line 259 in the former manuscript is now adapted to clarify this to the reader (please see response to reviewer 2 comment 3).

10. L268: See comment above about neglecting non-conservative processes in the free troposphere.

Please see specific comment 9. Line 268 in the former manuscript is adapted to clarify that non-conservative processes are neglected in the free troposphere for simplicity: "(these are neglected in the free troposphere for simplicity)."

11. L417: The role of latent heating in the free troposphere is largely ignored in this manuscript. The authors should further justify this neglection and try to quantify it.

Please see specific comment 9. This is clarified to the reader again at line 417:

"The differences between the two series is likely due to vertical and horizontal PV flux terms, but also possibly latent heating (which has not been considered here for simplicity)."

The focus of this study is the impact of the surface on Arctic cyclones. It is beyond the scope of this work to quantify the latent heat release in the free troposphere.

---

## Referee Report (RR1)

Review of wcd-2022-60_response

"The role of boundary layer processes in summer-time Arctic cyclones"
by
Hannah L. Croad, John Methven, Ben Harvey, Sarah P. E. Keeley, and Ambrogio Volonté

Recommendation: Minor revisions

General Comments:

The authors addressed several of the concerns raised, though some remain, see response to the authors' response below

> 1. … whilst we agree that a range of diagnostic tools are available to understand cyclone development, we feel that the PV framework provides a unique and valuable insight here…

As indicated, there have been other approaches to quantify the addressed effects, which confirms that the PV framework is not "unique" to this research question. Therefore, a comparison to these other findings using other methods was suggested and is still recommended, especially as some recent studies highly the importance of indirect effects of surface exchange, see comments further down.

> 3. … The central arguments in this work relate to thermal wind balance between the vorticity and potential temperature gradients, and what happens to cyclone structure when non-conservative processes modify either wind (friction) or potential temperature (diabatic processes). The benefit of using a PV framework is that structural changes within the cyclone can be inferred from the results. Changes in circulation and the constraint of thermal wind balance are not transparent in energetic frameworks and this is one of their major limitations.

PV only allows to assess structural changes related to stratification and vorticity if one implies balance assumptions, which, as pointed out, is highly questionable when focusing on processes in the boundary layer. The authors should further clarify how the use of PV should be enlightening in such a context, also given that they themselves state that "it is difficult to say how the BL PV tendencies contribute to the tropospheric depth-integrated circulation evolution." Furthermore, as pointed out further below, the neglect of how diabatic processes affect PV in the free troposphere, which is often argued to be rather significant for cyclone development, needs to be further substantiated.

> … the volume integral of the PV equation tells us about the processes contributing to changes in circulation without needing to invoke a specific balance relation.

While "a" circulation can be inferred, it is not given that it is "the" circulation associated with the circulation in the cyclone, e.g., if the tilt in the isentropes is

significant and the circulation is mainly occurring in the vertical plane. If assumptions about the stratification are needed to invoke inferences, this and potential sensitivities of the results should be clarified. The latter is also related to the response below to 5.

… this is not a precise interpretation of C when isentropic surfaces are tilted and intersect the upper or lower boundaries of the volume, as the reviewer points out. In the revised manuscript, we now use the more general term of "depth-integrated PV budget", rather than the "depth-integrated circulation budget", to ensure that the reader is not misled.

This is fine, though the authors' response above referred to circulation.

Whilst there may be latent heat release happening in the free troposphere, our study focuses on the effects of the surface on Arctic cyclones, including surface heat fluxes and frictional processes. Non-conservative processes in the free troposphere (including latent heating) are not examined and impact of the dry BL processes is isolated. … latent heat release is not examined for simplicity, so the impact of the dry BL processes is isolated.

As indicated in the original review, surface exchange can have direct and indirect effects on cyclone development, which have recently been assessed in both a PV and energy framework using theory and idealised numerical simulations, respectively (references see original review). These studies showed that the direct effects of surface exchange are usually small compared to the indirect effects (i.e., changes in latent heat release in the free troposphere). Hence, excluding free tropospheric non-conservative effects for "simplicity" for an investigation of surface effects on cyclones appears questionable. If it turns out that the non-conservative effects in the free troposphere are dominant, the exclusive focus of this study on only the direct effects of surface exchange could be misleading. It is also not correct to state that "the impact of the dry BL processes is isolated", as the indirect effects were neither controlled nor assessed.

If the inclusion of diabatic effects in the free troposphere is not feasible in the context of this study, the authors need to clearly state potential shortcomings of their study with respect to this neglect and how this might impact their main conclusions.

---

## Author Response (AR2)

**Author response: The role of boundary layer processes in summer-time Arctic cyclones**

The editor's and reviewer's comments are copied below in black, with our responses in blue. Note that all line numbers and figures refer to the previously revised copy of the manuscript that you evaluated.

**Editor:**

Thank you for your detailed revision and reply to the referees' comments. Based on the second round of reviews and my own evaluation, the manuscript may be acceptable for publication after some minor revision. Please address all the remaining concerns raised by Referee #3. For the revision, I also have some suggestions for consideration.

1. I agree with Referee 3 that it would be better to compare your results with previous studies using other methods, which will promote the contribution of your study to the topic.

Please see response to Reviewer 3, comment 1 and 5.

2. I also fully agree with the referee that the authors should add more discussions on the possible caveats of the study. As the referee suggested, the neglect of non-conservative processes in the free troposphere (including latent heating) is the one deserving discussion. Furthermore, I suggest the authors also made some discussions on the assumptions they made when including the BL processes in their PV budget calculation. For example, the formulas of the BL PV in Eqs. 9, 11,13,14 are all the ones derived for midlatitude cyclones. Are the boundary layer characteristics in Arctic similar to that in midlatitude for cyclones? How much do we know about the structure of the fluid in the Arctic BL? Why do the authors use such set of formulas? I suggest the authors either provide the rationale for using the formulas or adding some discussions in the conclusion part. Also, it is not clear to me how the authors calculate \tau_s and H_s in Eq. 8 in the study.

Regarding the caveats of the study (including neglect of non-conservative processes in the free troposphere), please see response to Reviewer 3, comment 5.

The motivation for using the BL PV tendency equations is that they have been used previously to investigate BL processes in mid-latitude cyclones (line 93-145), allowing direct comparison with these studies. The assumptions made in the derivation of the BL depth-averaged PV tendency equation (Equation 11) are only that the horizontal variation of fluxes is much smaller than the vertical variation (line 227-228), that there is a linear flux gradient in the BL (line 230), and that there is a constant density in the BL (line 238). These are general assumptions about turbulent mixing in the BL that are not specific to mid-latitudes or polar regions. To clarify this, a sentence is now added to the revised manuscript at line 248: "Although these equations have been derived and used previously in the context of mid-latitude cyclones, the assumptions made regarding turbulent mixing are equally applicable in the Arctic.".

Note that Equation 9 is the partition of the terms in Equation 6 into vertical (V) and horizontal (H) components, with no approximations made. In the newly revised manuscript we drop the V and H notation, which may be confusing to the reader, and re-write Equation 9 as follows:

$$\frac{DP}{Dt} = \underbrace{\frac{1}{\rho_0}\left[(\nabla \times \boldsymbol{F})\cdot\hat{\boldsymbol{k}}\frac{\partial\theta}{\partial z}\right]}_{(F_{EK})} + \underbrace{\frac{1}{\rho_0}\left[(\nabla \times \boldsymbol{F})\cdot\nabla_H\theta\right]}_{(F_{BG})} + \underbrace{\frac{1}{\rho_0}\left[\boldsymbol{\zeta_a}\cdot\hat{\boldsymbol{k}}\frac{\partial}{\partial z}\left(\frac{D\theta}{Dt}\right)_{shf}\right]}_{(S_V)}$$

$$+ \underbrace{\frac{1}{\rho_0}\left[\boldsymbol{\zeta_a}\cdot\nabla_H\left(\frac{D\theta}{Dt}\right)_{shf}\right]}_{(S_H)} + \underbrace{\frac{1}{\rho_0}\left[\boldsymbol{\zeta_a}\cdot\nabla\left(\frac{D\theta}{Dt}\right)_{lhf}\right]}_{(L)}$$

where $\nabla_H$ is the horizontal gradient operator.

The motivation for using the depth-integrated PV budget equations is to link the BL PV tendencies with cyclone evolution above the BL (line 262-263). The only assumptions made in the derivation of Equations 13 and 14 in the previously revised manuscript is that the BL height is flat (i.e. $\hat{n} = \hat{k}$). This was applied to simplify the expression for the flux across the BL top. However, as we don't calculate this term (i.e. it is not a major discussion point in the paper), in the newly revised manuscript we write the full equation without this assumption. Furthermore, in Equation 14 we have added the term representing non-conservative processes in the free troposphere (and explain in the manuscript that we do not calculate this term explicitly, as it is not the focus of this study). Equations 13 and 14 in the newly revised manuscript are as follows:

$$\frac{d}{dt}\langle P\rangle_{BL} = \iint \rho h \widehat{\frac{DP}{Dt}}\, dA - \iint_{z=h} \rho P(\boldsymbol{u}-\boldsymbol{u_b})\cdot\hat{\boldsymbol{n}}\, dA - \int_0^h \oint \rho P(\boldsymbol{u}-\boldsymbol{u_b})\cdot\hat{\boldsymbol{l}}\, dl dz$$

$$\frac{d}{dt}\langle P\rangle_{TROP} = \int_h^{z_{top}} \iint \rho\frac{DP}{Dt}\, dA dz + \iint_{z=h} \rho P(\boldsymbol{u}-\boldsymbol{u_b})\cdot\hat{\boldsymbol{n}}\, dA - \int_h^{z_{top}} \oint \rho P(\boldsymbol{u}-\boldsymbol{u_b})\cdot\hat{\boldsymbol{l}}\, dl dz$$

The surface momentum flux ($\tau_S$) and surface heat flux ($H_S$) are directly output from the ECMWF IFS model. This is now clarified to the reader in Section 2.2 by adding the following sentence at line 173: "The surface momentum flux and surface sensible heat flux are also used directly from the IFS model, and are computed using bulk formulae with exchange coefficients (ECMWF, 2020)."

3. Results in Figures 9 and 10 are interesting. I suggest the authors think of replotting some panels with different colors to make them easier to read. For example, it is very hard to distinguish curves for S_V and S_H under the yellow and purple background. In addition, in Figure 10a, why is L not at the center of the cyclone? How is the cyclone center defined?

We agree that Figures 9 and 10 could be improved to make them easier to read. In the revised manuscript, the background shading (denoting surface type) is now only displayed in a layer at the bottom of the subplots (rather than the full y-range). This reduces overlap with the curves and clashing of colours, so that the curves are now easier to distinguish from each other. We have also increased the thickness of the curves. We think that these revised plots are now easier for the reader to interpret.

In Figures 10 and 11 the vertical purple dashed line denotes the cyclone centre as determined by the TRACK algorithm, which is based on 850 hPa relative vorticity (see Section 2.3). The red L on these plots refers to the minimum sea level pressure across the section. The cyclone centre (based on 850 hPa relative vorticity) and minimum sea level pressure are not necessarily aligned, especially during the baroclinic growth phase (i.e. Cyclone B in Figure 10a). To remind the reader of how the cyclone centre is defined, a reference to Section 2.3 is now added in the caption for Figure 9 in the revised manuscript.

**Reviewer 3:**

The authors addressed several of the concerns raised, though some remain, see response to the authors' response below

1. "… whilst we agree that a range of diagnostic tools are available to understand cyclone development, we feel that the PV framework provides a unique and valuable insight here…"

As indicated, there have been other approaches to quantify the addressed effects, which confirms that the PV framework is not "unique" to this research question. Therefore, a comparison to these other findings using other methods was suggested and is still recommended, especially as some recent studies highly the importance of indirect effects of surface exchange, see comments further down.

We chose the PV framework to build on work by Cooper et al. (1992), Stoelinga (1996), Adamson et al. (2006) and others, but we acknowledge that other approaches could be used. Accordingly, we now refer to the results of Haualand and Spengler (2020) and Bui and Spengler (2021) in the introduction of the newly revised manuscript. However, as we haven't fully quantified the impact of friction and sensible heat fluxes on the cyclones in this study (i.e. quantified the response of the 3-D winds within the cyclones), we cannot put our results in context of the suggested studies here. Please see Reviewer 3, comment 5 for an extended discussion of this.

2. "… The central arguments in this work relate to thermal wind balance between the vorticity and potential temperature gradients, and what happens to cyclone structure when non-conservative processes modify either wind (friction) or potential temperature (diabatic processes). The benefit of using a PV framework is that structural changes within the cyclone can be inferred from the results. Changes in circulation and the constraint of thermal wind balance are not transparent in energetic frameworks and this is one of their major limitations."

PV only allows to assess structural changes related to stratification and vorticity if one implies balance assumptions, which, as pointed out, is highly questionable when focusing on processes in the boundary layer. The authors should further clarify how the use of PV should be enlightening in such a context, also given that they themselves state that "it is difficult to say how the BL PV tendencies contribute to the tropospheric depth-integrated circulation evolution." Furthermore, as pointed out further below, the neglect of how diabatic processes affect PV in the free troposphere, which is often argued to be rather significant for cyclone development, needs to be further substantiated.

In the derivation of the BL depth-averaged PV tendency equation, no assumptions about PV conservation or invertibility have been made, as stated at line 247-248 in the previously revised manuscript. The impact of friction and sensible heat fluxes on cyclones in model simulations has been demonstrated by other modelling studies using this PV framework (e.g.

Adamson et al., 2006; Boutle et al., 2007; Plant and Belcher, 2007). In this study no balance assumptions are needed to calculate or interpret the PV tendencies and their integrals (i.e. they are just diagnostic of the BL processes identified in previous modelling studies).

It is true that in order to relate the evolution of the cyclone outside the BL to the action of non-conservative processes within the BL, it is necessary to make approximations and to use a framework describing how the large-scale flow within a cyclone must respond to such changes. For example, in the simplest case of a barotropic cyclonic vortex, the action of friction within the BL on the vortex above the BL is typically explained using the Ekman pumping mechanism. The response of the secondary circulation in the r-z plane is central, but the mechanism can be understood qualitatively without detailed specification of a balance approximation, i.e. lines 104-111 in the previously revised manuscript, based on Hoskins and James (2014). This is the approach taken in this paper. The volume-integrated PV changes in the BL are partitioned into an Ekman term and other terms related to surface fluxes in a baroclinic environment. These terms are quantified explicitly. However, the response of the 3-D wind field to these terms is anticipated through a dynamical argument. In further work moving beyond this article, we are attempting to quantify the response of the winds (in a balanced-dynamics, semi-geotriptic, framework) to the non-conservative processes, but this involves a new tool and is beyond the scope of this article (see Reviewer 3, comment 5).

Here, we have qualitatively inferred the impacts of these processes by examining the 3-D structures of the cyclone. The dynamical arguments invoke thermal wind balance above the BL. In particular, we examine the $F_{BG}$ term and it's impact on the cyclone (summarised in Figure 11). We make the argument that the reduction of near-surface winds due to friction changes the wind shear across the BL. For winds just above the BL to remain in thermal wind balance, the horizontal temperature gradient (above the BL) must change accordingly (i.e. we are assuming thermal wind balance above the BL, not within the BL). To clarify this to the reader, we ensure that we refer to the thermal wind vector and temperature gradient "just above the BL" in the newly revised manuscript. It is apparent from the cross-sections in Figures 9b and 10b that thermal wind balance applies on the scale of the cyclone, with cyclone A having a warm-core and cyclonic flow maximum near the BL top, while cyclone B has a cold-core and the magnitude of the cyclonic flow increases with height.

Regarding diabatic processes in the free troposphere, please see response to Reviewer 3 comment 5.

3. "… the volume integral of the PV equation tells us about the processes contributing to changes in circulation without needing to invoke a specific balance relation."

While "a" circulation can be inferred, it is not given that it is "the" circulation associated with the circulation in the cyclone, e.g., if the tilt in the isentropes is significant and the circulation is mainly occurring in the vertical plane. If assumptions about the stratification are needed to invoke inferences, this and potential sensitivities of the results should be clarified. The latter is also related to the response below to 5.

We agree that the previous wording could have been misleading, and this has now been clarified to the reader in the previously revised manuscript at lines 265-269 (or lines 266-269 in the newly revised manuscript).

4. "… this is not a precise interpretation of C when isentropic surfaces are tilted and intersect the upper or lower boundaries of the volume, as the reviewer points out. In the revised manuscript, we now use the more general term of "depth-integrated PV budget", rather than the "depth-integrated circulation budget", to ensure that the reader is not misled."

This is fine, though the authors' response above referred to circulation.

All uses of the term "depth-integrated circulation" have been changed to "depth-integrated PV budget" in the revised manuscript.

5. "Whilst there may be latent heat release happening in the free troposphere, our study focuses on the effects of the surface on Arctic cyclones, including surface heat fluxes and frictional processes. Non-conservative processes in the free troposphere (including latent heating) are not examined and impact of the dry BL processes is isolated. … latent heat release is not examined for simplicity, so the impact of the dry BL processes is isolated."

As indicated in the original review, surface exchange can have direct and indirect effects on cyclone development, which have recently been assessed in both a PV and energy framework using theory and idealised numerical simulations, respectively (references see original review). These studies showed that the direct effects of surface exchange are usually small compared to the indirect effects (i.e., changes in latent heat release in the free troposphere). Hence, excluding free tropospheric non-conservative effects for "simplicity" for an investigation of surface effects on cyclones appears questionable. If it turns out that the non-conservative effects in the free troposphere are dominant, the exclusive focus of this study on only the direct effects of surface exchange could be misleading. It is also not correct to state that "the impact of the dry BL processes is isolated", as the indirect effects were neither controlled nor assessed.

If the inclusion of diabatic effects in the free troposphere is not feasible in the context of this study, the authors need to clearly state potential shortcomings of their study with respect to this neglect and how this might impact their main conclusions.

We re-write lines 134-138 to include reference to previous work on the impact of sensible heat fluxes on cyclones as follows: "Haualand and Spengler (2020) and Bui and Spengler (2021) demonstrated that the direct effect of surface sensible heat fluxes is to weaken mid-latitude cyclone development by reducing low-level baroclinicity, using PV and energy frameworks respectively in idealised modelling setups. However, both studies found that the impact of sensible heat fluxes was relatively small compared to that of latent heating. Sensible heat fluxes also modify the action of friction by altering BL stability and by weakening frontal gradients (Plant and Belcher, 2007)."

In this study we focus on characterising and understanding the effects of friction and sensible heat fluxes in the BL on two Arctic cyclone cases, and identifying the dependence of these frictional effects on large-scale cyclone structure. We have not attributed quantitatively the response of 3-D winds within the cyclones to these BL processes. This is a limitation and we suggest in the conclusions that this should be the subject of future work. Indeed, this is the subject of current work by the co-authors, including the role of latent heating (coupled with vertical motion). For this reason, we are not yet in the position to be able to put our results in the context of the indirect effects discussed by Haualand and Spengler (2020) and Bui and Spengler (2021). We also acknowledge that latent heating in the free troposphere may be a significant factor in the evolution of Arctic cyclones, but this is

not the subject of this particular study. To clarify this to the reader, we have made the following changes to the manuscript:

- Equation 14 has been re-written to include a term representing non-conservative processes in the free troposphere. Rather than this term being "neglected", we explain in Section 2.6 that this term is not calculated explicitly in this work (along with the vertical and horizontal fluxes of PV on the RHS of the equation), as the action of non-conservative processes in the free troposphere are not the subject of this study. However, the changes in $\langle P \rangle$ diagnosed in the IFS model include the effects of all processes including latent heat release above the BL.
- In particular, lines 279-281 have been re-written as "Non-conservative processes in the BL and free troposphere are included in the formulation (although the latter are not calculated explicitly). Note that whilst non-conservative processes in the free troposphere may occur at mid-levels within Arctic cyclones, in particular latent heating, these are not the subject of this study. However, the changes in $\langle P \rangle$ diagnosed in the IFS model include the effects of all processes, including latent heat release above the BL.".
- The brackets on line 453 are re-written as "(which is not explicitly calculated here)".
- We extend the future work section at the end of the conclusions (lines 654-658): "Moist processes and diabatic effects in the free troposphere (in particular latent heat release coupled with the vertical motion) have not been considered here. Although we may expect latent heating to be less important in the Arctic than lower latitudes due to reduced absolute humidity, Terpstra et al. (2015) demonstrated that low-level disturbances are able to amplify in a high-latitude moist baroclinic environment in the absence of other processes (upper-level perturbations, surface fluxes or radiation) using an idealised baroclinic channel model. This suggests that latent heating can be significant for the development of polar cyclones. The work here is focused on characterising the effects of friction and sensible heat fluxes at the lower boundary on Arctic cyclones in two cases with contrasting structure. The authors are conducting further study to quantify the response of the 3-D wind field within the cyclones to the BL processes explored here, and the amplification of ascent by latent heat release, using the diagnostic tool of Cullen (2018) assuming semi-geostrophic balance dynamics. Quantifying the relative importance of non-conservative processes in the BL and free troposphere in the evolution of Arctic cyclones and understanding the sensitivity of cyclone evolution to surface properties, are also areas for future research.".

References added to manuscript:

Bui, H., and Spengler, T.: On the Influence of Sea Surface Temperature Distributions on the Development of Extratropical Cyclones. *J. Atmos. Sci.*, **78**, 1173-1188, 2021.

Cullen, M.: The use of Semigeostrophic Theory to Diagnose the Behaviour of an Atmospheric GCM, *Fluids*, **3**, 72, 2018.

Terpstra, A., Spengler, T., and Moore, R. W.: Idealised simulations of polar low development in an Arctic moist-baroclinic environment. *Quart. J. Roy. Meteor. Soc.*, **141**, 1987-1996, 2015.